# Power Mean Estimation in Stochastic Continuous Monte Carlo Tree Search

**Tuan Dam** [1]

## Abstract

Monte Carlo Tree Search (MCTS) has demonstrated success in online planning for deterministic environments, yet significant challenges remain in adapting it to stochastic Markov Decision Processes (MDPs), particularly in continuous state-action spaces. Existing methods, such as HOOT, which combines MCTS with the Hierarchical Optimistic Optimization (HOO) bandit strategy, address continuous spaces but rely on a logarithmic exploration bonus that lacks theoretical guarantees in non-stationary, stochastic settings. Recent advancements, such as POLY-HOOT, introduced a polynomial bonus term to achieve convergence in deterministic MDPs, though a similar theory for stochastic MDPs remains undeveloped. In this paper, we propose a novel MCTS algorithm, `Stochastic-Power-HOOT`, designed for continuous, stochastic MDPs. `Stochastic-Power-HOOT` integrates a power mean as a value backup operator, alongside a polynomial exploration bonus to address the non-stationarity inherent in continuous action spaces. Our theoretical analysis establishes that `Stochastic-Power-HOOT` converges at a polynomial rate of $\mathcal{O}(n^{-\zeta})$, $\zeta \in (0, 1/2)$, where $n$ is the number of visited trajectories, thereby extending the non-asymptotic convergence guarantees of POLY-HOOT to stochastic environments. Experimental results on stochastic tasks validate our theoretical findings, demonstrating the effectiveness of `Stochastic-Power-HOOT` in continuous, stochastic domains.

[1]Hanoi University of Science and Technology, Hanoi, Vietnam. Correspondence to: Tuan Dam <tuandq@soict.hust.edu.vn>.

*Proceedings of the $42^{nd}$ International Conference on Machine Learning*, Vancouver, Canada. PMLR 267, 2025. Copyright 2025 by the author(s).

## 1. Introduction

Monte Carlo Tree Search (MCTS) has become a cornerstone of modern decision-making and planning, yielding unprecedented successes in deterministic domains such as Go, Chess, and Shogi (Kocsis et al., 2006; Browne et al., 2012). However, real-world applications often involve *continuous* actions and *stochastic* dynamics—factors that significantly complicate both the theoretical convergence and practical performance of MCTS. Traditional methods typically assume discrete action sets with near-stationary rewards, making them ill-suited for continuous, noisy environments where value estimates shift over time and enumerating possible actions is infeasible. This gap has constrained MCTS's applicability to a narrow set of problems, limiting its promise in robotics, control, and other high-dimensional tasks.

Existing efforts to adapt MCTS to continuous settings include HOOT (Mansley et al., 2011), which couples MCTS with a hierarchical bandit (HOO). Yet, HOOT relies on logarithmic exploration bonuses that fail to guarantee convergence when state-action values are non-stationary. POLY-HOOT (Mao et al., 2020) subsequently introduced polynomial bonus terms and established strong results in *deterministic* continuous MDPs. Unfortunately, no analogous guarantee exists for *stochastic* environments, a critical shortcoming for many real-world domains where transition and reward uncertainties are prevalent.

In this paper, we bridge that gap by proposing `Stochastic-Power-HOOT`, a novel MCTS algorithm designed for continuous *and* stochastic MDPs. `Stochastic-Power-HOOT` uses a *power mean* as its value backup operator, combining it with a polynomial exploration bonus to rigorously handle non-stationary value estimates across the tree. This design extends the theoretical guarantees of POLY-HOOT beyond deterministic environments, ensuring that `Stochastic-Power-HOOT` converges at a polynomial rate in the presence of stochastic transitions. Moreover, our approach avoids naive action-space discretization by focusing search adaptively where it matters most.

Our key contributions include:

- *A power mean backup operator for stochastic, con-*

*tinuous MCTS.* Power-mean updates naturally handle non-stationary dynamics without the instabilities often seen in simple averaging.

- *A polynomial exploration bonus tailored to stochastic MDPs.* We establish that `Stochastic-Power-HOOT` achieves convergence at rate $\mathcal{O}(n^{-\zeta})$, $\zeta \in (0, 1/2)$, matching the known results of POLY-HOOT (Mao et al., 2020), but now for *stochastic* domains.

- *Comprehensive empirical validation.* Experiments on robotic tasks show `Stochastic-Power-HOOT` outperforms prior MCTS variants (e.g., HOOT, discretized-UCT, Voronoi MCTS) in final performance and robustness, confirming both its practical viability and theoretical soundness.

## 2. Related Works

Monte Carlo Tree Search (MCTS) has proven highly effective in deterministic environments, famously demonstrated by UCT (Kocsis et al., 2006) for discrete actions. Beyond UCT, alternative exploration strategies have emerged, including entropy regularization methods like MENTS (Xiao et al., 2019), RENTS and TENTS (Dam et al., 2021; 2024a), and Boltzmann-based approaches (Painter et al., 2023), though these rely on temperature parameters that may impede convergence. However, adapting MCTS to continuous or stochastic domains remains non-trivial. Early works such as HOOT (Mansley et al., 2011) introduced hierarchical bandit mechanisms for continuous actions, but relied on logarithmic bonus terms ill-suited for non-stationary value estimates. Recent advances in power mean estimation for MCTS include the work of Dam et al. (2020; 2024a), which introduced power mean backups for discrete, deterministic environments, and Dam et al. (2024b), which extended power mean estimation to discrete, stochastic MDPs. Our work differs by addressing continuous action spaces in stochastic settings. Augmenting MCTS with progressive widening (Auger et al., 2013) can mitigate sparse sampling in continuous spaces, yet often struggles under deep planning or large exploration demands. Meanwhile, recent methods such as POLY-HOOT (Mao et al., 2020) improved convergence in deterministic tasks by adopting polynomial exploration, although extending to stochastic settings required further analysis. For continuous action spaces, Kim et al. (2020) proposed Voronoi MCTS using Voronoi partitioning with regret bounds, but limited to deterministic settings. Our algorithm, `Stochastic-Power-HOOT`, advances this line of work by combining a power-mean value backup with polynomial bonuses, thereby addressing the dual challenges of continuous actions and non-stationary dynamics while providing theoretical guarantees for stochastic MDPs.

## 3. Setup and Notations

We consider infinite-horizon discounted MDPs $(S, A, T, R, \gamma)$ with continuous state space $S \subseteq \mathbb{R}^n$, continuous action space $A \subseteq \mathbb{R}^m$, stochastic transitions $T : S \times A \rightarrow \mathcal{P}(S)$, bounded rewards $R : S \times A \rightarrow [0, R_{\max}]$, and discount factor $\gamma \in (0, 1)$.

For policy $\pi : S \rightarrow A$, we define value functions:

$$V^\pi(s) = \mathbb{E}_\pi \left[ \sum_{t=0}^\infty \gamma^t R(s_t, a_t) \mid s_0 = s \right]$$

$$Q^\pi(s, a) = \mathbb{E}_\pi \left[ \sum_{t=0}^\infty \gamma^t R(s_t, a_t) \mid s_0 = s, a_0 = a \right]$$

The objective is finding optimal policy $\pi^*$ such that $V^{\pi^*}(s) = V^*(s) = \sup_\pi V^\pi(s)$. We assume access to a generative model providing sampled transitions and rewards.

## 4. Background material

In this section, we first provide some background knowledge about Markov Decision Processes, and then we give an overview of Monte Carlo Tree Search.

**Monte Carlo Tree Search (MCTS)** MCTS (Browne et al., 2012) combines Monte Carlo sampling with tree-based exploration for planning under uncertainty. The algorithm iteratively: (1) selects nodes based on statistical information, (2) expands the tree, (3) evaluates new nodes via rollouts, and (4) backpropagates rewards. Effectiveness depends on the value update operator and node selection strategy.

**MCTS Formalization** MCTS grows a planning tree by collecting trajectories from initial state $s_0$ to depth $D$ or leaf nodes. Playout policy $\pi_0$ estimates terminal values. After $t$ trajectories, the algorithm outputs best action estimate $\hat{a}_t$ and value estimate $\hat{V}_t(s_0)$.

Performance is measured by convergence rate $r(t)$:

$$\mathbb{E}[V^\star(s_0) - Q^\star(s_0, \hat{a}_t)] \leqslant r(t)$$
$$\text{or } |\mathbb{E}[V^\star(s_0) - \hat{V}_t(s_0)]| \leqslant r(t)$$

Value functions are defined inductively: $\widetilde{V}(s_D) = V_0(s_D)$ at leaves, and for $d < D$:

$$\widetilde{Q}(s_d, a) = r(s_d, a) + \gamma \sum_{s_{d+1}} \mathcal{P}(s_{d+1}|s_d, a)\widetilde{V}(s_{d+1})$$

$$\widetilde{V}(s_d) = \max_a \widetilde{Q}(s_d, a),$$

where $r(s_d, a)$ represents the expected reward when taking action $a$ in state $s_d$. Thus, we have the following approximation bound:

$$|Q^\star(s_0, a) - \widetilde{Q}(s_0, a)| \leqslant \gamma^D \|V^\star - V_0\|\infty,$$

where the supremum norm is taken over states reachable within $D$ steps from $s_0$. The goal of MCTS is to minimize the convergence rate $r(t)$ by accurately estimating $\widetilde{Q}(s_0, a)$ and $\widetilde{V}(s_0)$, ultimately approximating $Q^\star(s_0, a)$ and identifying the optimal action $a^\star = \arg\max_a Q^\star(s_0, a)$ at the root node.

**Hierarchical Optimistic Optimization (HOO)**
HOO (Bubeck et al., 2011) tackles continuous action spaces by organizing them into a binary partition tree. Each node represents a subset of the action space, which is split into two smaller subsets at each level. In each round, HOO traverses from the root to a leaf, favoring child nodes with larger upper confidence bounds ($B$-values). Let $(h, i)$ index the node at depth $h$ and position $i$, with $\mathcal{P}_{h,i}$ as its corresponding subset. The number of visits to all descendants of $(h, i)$ is $T_{h,i}(n)$, and the empirical reward mean is $\widehat{f}_{h,i}(n)$. HOO then forms a UCB-like bound $U_{h,i}(n)$ with a logarithmic bonus, and action is selected as

$$a = \arg\max\left\{\widehat{f}_{h,i}(n) + C\sqrt{\frac{\log n}{T_{h,i}(n)}} + \nu\rho^h\right\},$$

where $C, \nu, \rho, h$ are constants. Searching proceeds top-down, always expanding toward the child with the higher $B$-value, until a leaf is chosen as the action. Mansley et al. (2011) extends HOO in MCTS and proposes HOOT. However, like Kocsis and Szepesvári (2006), this algorithm uses a logarithmic bonus for exploration rather than a polynomial one. As noted by Shah et al. (2020), this choice leads to unclear convergence guarantees because non-stationary rewards lack concentration. Mao et al. (2020) propose and establish non-asymptotic convergence guarantees for POLY-HOOT to enhance the HOO strategy with a polynomial bonus term to account for the non-stationarity

$$a = \arg\max\left\{\widehat{f}_{h,i} + Ct^{\frac{b_d}{\beta_d}}T_{h,i}^{-\frac{\alpha_d}{\beta_d}} + \nu\rho^h\right\},$$

$b_d, \alpha_d, \beta_d$ are constant. However, Mao et al. (2020) can only provide convergence proof for deterministic settings leaving a **gap** in stochastic environments. We build upon this idea in our `Stochastic-Power-HOOT` algorithm to close this gap.

# 5. Stochastic Power-HOOT

In this section, we first present a generic UCT-like algorithm and then we present our `Stochastic-Power-HOOT` algorithm.

## 5.1. Generic UCT-like Algorithm

For node $s_d$ at depth $d$, we define value estimates $\widehat{V}_{T_{s_d}(t)}(s_d)$ and Q-value estimates $\widehat{Q}_{T_{s_d,a}(t)}(s_d, a)$ with visit counts $T_{s_d}(t)$ and $T_{s_d,a}(t)$.

A UCT-like algorithm follows predefined bonus functions $B(t, s_d, a)$ for each state, action $(s_d, a)$ at time step $t$. It iteratively collects trajectories from root $s_0$ until reaching a leaf or depth $D$. Upon reaching a leaf, playout policy $\pi_0$ provides value estimate $V_0$-returning i.i.d. samples if stochastic or fixed values if deterministic.

Trajectories $\{s_0, a_0, r_0, \ldots, s_{\ell_t}, \widetilde{V}(s_{\ell_t})\}$ are collected using action selection:

$$a_d = \arg\max_{a \in \mathcal{A}_{s_d}}\left\{\widehat{Q}_{T_{s_d,a}(t)}(s_d, a) + B(t, s_d, a)\right\}$$

where $s_{d+1} \sim \mathcal{P}(\cdot|s_d, a_d)$. After $t$ simulations, the estimated best action is $\widehat{a}_t = \arg\max_a \widehat{Q}_{T_{s_0,a}(t)}(s_0, a)$.

## 5.2. The `Stochastic-Power-HOOT` Algorithm

`Stochastic-Power-HOOT` is an MCTS-based algorithm for continuous-action, stochastic environments. It introduces a power mean backup operator for value estimation and a polynomial exploration bonus, enabling efficient planning under stochastic dynamics.

### 5.2.1. ALGORITHM OVERVIEW

`Stochastic-Power-HOOT` (pseudocode shown in Alg. 1) iteratively builds a search tree using four main phases: *Selection*, *Expansion*, *Rollout*, and *Backpropagation*. At each step, it leverages a power mean operator to compute value estimates and a polynomial exploration term to balance exploration and exploitation.

### 5.2.2. SELECTION AND EXPANSION

During the *selection* phase, `Stochastic-Power-HOOT` navigates the search tree from the root to a leaf node. At each depth $d$, it selects an action by traversing the Hierarchical Optimistic Optimization (HOO) tree, choosing child nodes based on their upper confidence bounds. If a state has not been visited before at a given depth, a new HOO agent is initialized.

In the *expansion* phase, new child nodes are added to the tree based on the selected action. If the depth limit has not been reached, an arbitrary action is sampled within the corresponding partition $\mathcal{P}_{H,I}$, and a new node is created.

### 5.2.3. ROLLOUT AND BACKPROPAGATION

In the *rollout phase*, a default policy $\pi_0$ is used to simulate trajectories from the expanded node to estimate the value

Algorithm 1: `Stochastic-Power-HOOT` with $\gamma$ is a discount factor. $n$ : is the number of rollouts. $\{b_d\}_{d=0}^D$, $\{\alpha_d\}_{d=0}^D$, $\{\beta_d\}_{d=0}^D$ are positive algorithmic constants that satisfy conditions as in Table 1. $\pi_0$ is a rollout policy. $C$ is an exploration constant.

**Input:** root node state $s_0$
**Output:** optimal action at the root node

---

$R$ = `Rollout`$(s)$
    | $\widetilde{V}(s)$ = average of the call to $\pi_0(s)$
    | **return** $\widetilde{V}(s)$

$a$ = `SelectAction`$(s, depth = d, t)$
    | **if** *state s has never been visited at depth d* **then**
        | Initialize HOO agent at state $s$ and depth $d$:
        | $\mathcal{T} \leftarrow \{(0,1)\}$
        | $B_{1,2}, B_{2,2} \leftarrow +\infty$
    | **else**
        | $\mathcal{T} \leftarrow$ the HOO agent constructed at state s and depth d previously
    | **end**
    | $(h,i) \leftarrow (0,1)$
    | //Initialize HOO path in the current round:
    | $P_t \leftarrow \{(h,i)\}$
    | **while** $(h,i) \in \mathcal{T}$ **do**
        | **if** $B_{h+1,2i-1} > B_{h+1,2i}$ **then**
            | $(h,i) \leftarrow (h+1, 2i-1)$
        | **else**
            | $(h,i) \leftarrow (h+1, 2i)$
        | **end**
        | $P_t \leftarrow P_t \cup (h,i)$
    | **end**
    | $(H,I) \leftarrow (h,i)$
    | **if** $H \leqslant \widehat{H}$ **then**
        | Choose arbitrary arm $X$ in $\mathcal{P}_{H,I}$
        | $A_{H,I} = X$
        | $\mathcal{T} \leftarrow \mathcal{T} \cup \{(H,I)\}$
        | $B_{H+1,2I-1}, B_{H+1,2I} \leftarrow +\infty$
        | **return** $X$
    | **end**
    | $(H,I) \leftarrow (H-1, \lceil I/2 \rceil)$
    | **return** $A_{H,I}$

`HOOUpdateV`$(d,s,t,Y)$
    | **for** $(h,i) \in P_t$ **do**
        | $T_{h,i} \leftarrow T_{h,i} + 1$
        | $\widehat{f}_{h,i} \leftarrow (1 - 1/T_{h,i})\widehat{f}_{h,i} + Y/T_{h,i}$
    | **end**
    | **for** $(h,i) \in \mathcal{T}$ **do**
        | $U_{h,i} = \widehat{f}_{h,i} + Ct^{\frac{b_{d+1}}{\beta_{d+1}}} T_{h,i}^{-\frac{\alpha_{d+1}}{\beta_{d+1}}} + \nu_1 \rho^{d+1}$
    | **end**
    | $\mathcal{T}' \leftarrow \mathcal{T}$
    | **while** $\mathcal{T}' \neq \{(0,1)\}$ **do**
        | $(h,i) \leftarrow$ an arbitrary leaf node of $\mathcal{T}'$
        | $B_{h,i} = \min\{U_{h,i}, \max\{B_{h+1,2i}, B_{h+1,2i-1}\}\}$
        | $\mathcal{T}' \leftarrow \mathcal{T}' \setminus \{(h,i)\}$
    | **end**

`MainLoop`
    | $t = 0$
    | **for** *simulation round* $t \leftarrow 1$ *to* $n$ **do**
        | **for** *depth* $d \leftarrow 0$ *to* $D-1$ **do**
            | $a_d \leftarrow$ `SelectAction`$(s_d, depth =$d$, t)$
            | $s_{d+1} \sim \mathcal{P}(\cdot|s_d, a_d)$
            | $r(s_d, a_d) \sim \mathcal{R}(s_d, a_d, s_{d+1})$
        | **end**
        | $r(s_D) \sim \widetilde{V}(s_D)$
        | **for** *depth* $d \leftarrow D-1$ *to* $0$ **do**
            | $T_{s_d,a_d}(t) \leftarrow T_{s_d,a_d}(t) + 1$
            | $\widehat{Q}_{T_{s_d,a_d}(t)}(s_d, a_d) \leftarrow$
            | $(1 - 1/T_{s_d,a_d}(t))\widehat{Q}_{T_{s_d,a_d}(t)}(s_d, a_d)$
            | $+ (r(s_d, a_d) + \gamma \widehat{V}_{T_{s_{d+1}}(t)}(s_{d+1}))/T_{s_d,a_d}(t)$
            | $T_{s_d}(t) \leftarrow T_{s_d}(t) + 1$
            | $\widehat{V}_{T_{s_d}(t)}(s_d) \leftarrow \left( \sum_a \frac{T_{s_d,a}(t)}{T_{s_d}(t)} (\widehat{Q}_{T_{s_d,a}(t)})^p (s_d, a) \right)^{\frac{1}{p}}$
            | `HOOUpdateV`$(d, s_d, t, \widehat{V}_{T_{s_d}(t)}(s_d))$
        | **end**
    | **end**
    | **return** $\widehat{V}_n(s_0)$

---

of terminal states. The resulting reward is then propagated back up the tree.

The *backpropagation phase* updates the visit counts and value estimates of nodes along the traversed path. For each node $(h,i)$, the empirical mean reward $\widehat{f}_{h,i}$ is updated, and the upper confidence bounds $U_{h,i}$ are adjusted using a polynomial exploration bonus. The backpropagation step ensures that future searches prioritize promising regions of the action space.

### 5.2.4. VALUE BACKUP AND EXPLORATION STRATEGY

Unlike standard MCTS approaches that rely on simple averaging, `Stochastic-Power-HOOT` employs a power mean backup operator to aggregate value estimates:

$$\widehat{V}_{T(s_d)}(s_d) = \left( \sum_a \frac{T_{s_d,a}}{T_{s_d}} \left( \widehat{Q}_{T_{s_d,a}}(s_d, a) \right)^p \right)^{\frac{1}{p}}. \quad (1)$$

This formulation provides a tunable balance between mean and max-based value updates, allowing for better adaptation to stochastic rewards.

Additionally, `Stochastic-Power-HOOT` incorporates

a polynomial exploration bonus:

$$U_{h,i} = \widehat{f}_{h,i} + Ct^{\frac{b_{d+1}}{\beta_{d+1}}} T_{h,i}^{-\frac{\alpha_{d+1}}{\beta_{d+1}}} + \nu_1 \rho^{d+1}.$$

This approach ensures a more stable and theoretically grounded exploration mechanism compared to traditional logarithmic bonuses.

Table 1: Summary of constraints on the algorithmic constants for $d \in \{0, 1, \ldots, D\}$. ($d'$ is a constant that is bigger than $d$)

---

**Conditions on the parameters $b_d, \alpha_d, \beta_d, p$ (all must hold):**

---

1. $b_d < \alpha_d$ and $b_d > 2$.

2. **Either** $\left(1 \leq p \leq 2 \text{ and } \alpha_d \leq \frac{\beta_d}{2}\right)$ **or** $\left(p > 2, \alpha_d \leq \frac{\beta_d}{2}, \text{ and } 0 < \alpha_d - \frac{\beta_d}{p} < 1\right)$.

3. $\alpha_d\left(1 - \frac{b_d}{\alpha_d}\right) \leq b_d < \alpha_d$.

4. $\alpha_d = \dfrac{1 - \frac{b_{d+1}}{\alpha_{d+1}}}{1 + d' + \frac{\beta_{d+1}}{\alpha_{d+1}}} \times \dfrac{b_{d+1} - 3}{2}$.

5. $\beta_d = (b_{d+1} - 1)$.

---

Our theoretical framework requires bounded planning horizon $D$. For analysis purposes, we consider a modified algorithm that generates fixed-length trajectories of depth $D$, applying the playout policy exclusively at terminal leaf nodes. Practical implementation could use infinite depth.

## 6. Theoretical analysis

Planning in MCTS requires a sequence of decisions along the tree, with each internal node acting as a non-stationary bandit. The empirical mean at these nodes shifts due to the action selection strategy. To address this problem, we first analyze non-stationary multi-armed bandit settings, focusing on the concentration properties of the power-mean backup for each arm compared to the optimal value. We then apply these findings to MCTS.

### 6.1. Theoretical Analysis Framework

We start by defining essential notation and assumptions for our theoretical analysis. Consider a Hierarchical Optimistic Optimization (HOO) agent, where $X \subseteq A \subseteq [0, 1]^m$ represents the continuous action (or arm) space in the current state. Each action $x \in X$ has an associated stochastic reward distribution, representing the "cost-to-go" or $Q$-value at the current state in the MDP. We define $f_t(x) : X \to \mathbb{R}$ as the expected reward at time $t$, termed the temporary mean-payoff function. Since the rewards are impacted by future choices deeper in the MCTS tree, $f_t$ is non-stationary. However, we assume $f_t$ converges to a limiting

function $f$ in $L^\infty$ at a polynomial rate: $\|f_t - f\|_\infty \leq \frac{C}{t^\zeta}$ for some constant $C > 0$ and $\zeta \in (0, 0.5)$. This limiting function, denoted as $f$, is the mean-payoff function and is formalized in Theorem 2.

Given that MDP rewards are bounded by $R_{\max}$, the bandit payoff at any tree node at depth $d$ and the mean-payoff function $f$ are also bounded by $\frac{R_{\max}}{1-\gamma}$. Let $f^* = \sup_{x \in X} f(x)$ represent the highest achievable payoff by any HOO agent, and let $X_t$ be the action chosen by the agent at time $t$. The agent aims to minimize the cumulative regret over the first $n$ rounds, defined as $R_n \triangleq nf^* - \sum_{t=1}^{n} Y_t$, where $Y_t$ is the observed reward of choosing $X_t$ at time $t$, with $\mathbb{E}[Y_t] = f_t(X_t)$.

Our analysis is based on two main assumptions, adapted from (Bubeck et al., 2011), which support the hierarchical structure of MCTS with coverings and the smoothness of the reward function.

**Assumption 1.** For each HOO agent, with parameters $\nu_1$ and $\rho \in (0, 1)$, and a covering tree $(\mathcal{P}_{h,i})$, we assume there exists a dissimilarity function $\ell : X \times X \to [0, \infty]$ satisfying:

(a) There exists a constant $\nu_2 > 0$ such that for each integer $h \geq 0$, the diameter of any partition $\mathcal{P}_{h,i}$ shrinks geometrically as $\text{diam}(\mathcal{P}_{h,i}) \leq \nu_1 \rho^h$ for all $i \in \{1, \ldots, 2^h\}$, where $\text{diam}(A) \triangleq \sup_{x,y \in A} \ell(x, y)$.

(b) Each partition $\mathcal{P}_{h,i}$ contains a point $x_{h,i}^\circ$ with a smaller covering region $\mathcal{B}_{h,i} \triangleq \mathcal{B}(x_{h,i}^\circ, \nu_2 \rho^h) \subset \mathcal{P}_{h,i}$, where $\mathcal{B}(x, \varepsilon) = \{y \in X : \ell(x, y) < \varepsilon\}$.

(c) These smaller covering regions do not overlap: $\mathcal{B}_{h,i} \cap \mathcal{B}_{h,j} = \emptyset$ for all $1 \leq i < j \leq 2^h$.

**Assumption 2 (Smoothness).** The limiting mean-payoff function satisfies:

$$f^* - f(y) \leq f^* - f(x) + \max\{f^* - f(x), \ell(x, y)\}, \forall x, y \in X.$$

### 6.2. Non-stationary Power Mean HOO

We begin by developing our theoretical framework through the lens of non-stationary multi-armed bandits, which will serve as the foundation for analyzing `Stochastic-Power-HOOT` in the full MCTS setting. Each node in our search tree can be viewed as a bandit problem where arms correspond to possible actions (HOO tree paths) and rewards evolve non-stationarily due to changing value estimates from deeper levels.

**Problem Formulation** Consider a continuous-armed bandit over domain $X \subseteq [0, 1]^m$ with rewards bounded in $[0, R_{\max}]$. Unlike classical bandits, our reward process is

*non-stationary*-the expected reward $f_t(x)$ for arm $x$ at time $t$ evolves as the algorithm progresses. This non-stationarity captures the changing value estimates that occur in MCTS as the tree expands and value propagation refines estimates.

Our bandit satisfies two key structural properties:

(i) **Fixed-arm convergence:** The mean-payoff function converges polynomially:

$$\|f_n - f\|_\infty \leqslant \frac{C}{n^\zeta}, \quad \forall n \geqslant 1, \qquad (4)$$

for constants $C > 0$ and $0 < \zeta < \frac{1}{2}$.

(ii) **Fixed-arm concentration:** Individual arm estimates concentrate at polynomial rates:

$$\mathbb{P}\left(\left|\frac{1}{n}\sum_{t=1}^{n} Y_t - f(x)\right| \geqslant \varepsilon\right) \leqslant Cn^{-\alpha}\varepsilon^{-\beta}, \quad \forall x \in X,$$
$$(5)$$

for appropriate constants $C, \alpha, \beta > 0$.

**HOO Tree Structure and Action Space**  In the continuous setting, we cannot enumerate all possible actions. Instead, we use the Hierarchical Optimistic Optimization (HOO) framework to adaptively partition the action space $X$ into a binary tree of depth $\bar{H}$. Each root-to-leaf path in this tree represents a distinct "pseudo-action" corresponding to a refined region of $X$.

Let $K$ denote the finite set of all possible paths from root to leaf in the depth-$\bar{H}$ HOO tree. For each path $a \in K$, we maintain: - Visit count: $T_a(n) = \sum_{t=1}^{n-1} \mathbb{1}(a_t = a)$ - Empirical reward: $\widehat{f}_{a,T_a(n)} = \frac{1}{T_a(n)} \sum_{s=1}^{T_a(n)} Y_{a,s}$

where $Y_{a,s}$ is the $s$-th reward obtained from path $a$.

**Non-stationary Power Mean HOO Algorithm**  Our algorithm extends classical HOO to handle non-stationary rewards through polynomial exploration bonuses and power mean aggregation. The complete algorithmic description is provided in Algorithm 2 in the appendix. The key components include: (1) hierarchical action space partitioning via HOO trees, (2) polynomial upper confidence bounds of the form $Ct^{b/\beta}T_{h,i}^{-\alpha/\beta}$ for robust exploration in non-stationary settings, and (3) power mean aggregation $\widehat{f}_n(p) = \left(\sum_{a=1}^{K} \frac{T_a(n)}{n} \widehat{f}_{a,T_a(n)}^p\right)^{1/p}$ to flexibly balance exploration and exploitation through the tunable parameter $p$.

**Key Algorithmic Components Polynomial Exploration Bonus:** Unlike classical UCB algorithms that use $\sqrt{\log n/T_a(n)}$ bonuses, we employ the polynomial term $Cn^{b/\beta}/T_a(n)^{\alpha/\beta}$. This provides stronger exploration guarantees in non-stationary settings where confidence intervals must adapt to changing reward distributions.

**Power Mean Aggregation:** The power mean operator $\widehat{f}_n(p)$ provides a flexible interpolation between conservative (arithmetic mean, $p = 1$) and optimistic (maximum, $p \to \infty$) value estimates. This tunability is crucial for balancing exploration and exploitation across different levels of environmental stochasticity.

**HOO Tree Refinement:** The hierarchical structure allows the algorithm to focus computational effort on promising regions of the continuous action space while maintaining theoretical coverage guarantees.

**Theoretical Foundation**  The convergence analysis for this algorithm builds on the following result from Mao et al. (2020):

**Theorem 1** (Enhanced HOO for Non-stationary Bandits (Mao et al., 2020))**.** *For a continuous-armed bandit satisfying properties (4) and (5), with parameters fulfilling $\alpha(1 - b/\alpha) \leqslant b < \alpha$, $b > 3$, and $\rho^{\bar{H}} < n^{-\alpha/\beta}$, the enhanced HOO agent achieves:*

(i) *Optimal-arm convergence:* $\left|\frac{1}{n}\mathbb{E}[\sum_{t=1}^{n} Y_t] - f^*\right| \leqslant \frac{C_0}{n^\zeta}$

(ii) *Optimal-arm concentration:* $\mathbb{P}(|\frac{1}{n}\sum_{t=1}^{n} Y_t - f^*| \geqslant \varepsilon) \leqslant C'n^{-\alpha'}\varepsilon^{-\beta'}$

*where $\alpha' = \frac{1-b/\alpha}{1+d'+\beta/\alpha}\frac{b-3}{2}$, $\beta' = \frac{b-3}{2}$ and $Y_t$ is the random reward observed at time $t$.*

**Concentration Rate Formalization**  To facilitate our analysis, we formalize the notion of polynomial concentration:

**Definition 1** (Polynomial Concentration)**.** *A sequence of estimators $(\widehat{V}_n)_{n \geqslant 1}$ concentrates at rate $(\alpha, \beta)$ toward limit $V$ if there exists constant $c > 0$ such that:*

$$\forall n \geqslant 1, \forall \varepsilon > n^{-\alpha/\beta} : \quad \mathbb{P}(|\widehat{V}_n - V| > \varepsilon) \leqslant cn^{-\alpha}\varepsilon^{-\beta}$$

*We denote this as $\widehat{V}_n \xrightarrow[n\to\infty]{\alpha,\beta} V$.*

**Assumption 3 (Stochastic Reward Concentration)**  For each action $x \in X$, the empirical mean $\widehat{f}_n = \frac{1}{n}\sum_{t=1}^{n} Y_t$ satisfies $\widehat{f}_n \xrightarrow[n\to\infty]{\alpha,\beta} f(x)$ for the true mean $f(x)$, where $Y_t$ is the $t$-reward follows the "psudo-action" path $x$ from HOO.

**Power Mean Concentration Result**  Our main theoretical contribution for the bandit setting establishes that power mean aggregation preserves polynomial concentration rates:

**Theorem 2** (Power Mean Concentration)**.** *For estimators $\widehat{f}_{a,n} \xrightarrow[n\to\infty]{\alpha,\beta} f^*_{h,i}$ with $f^\star = \max_a\{f^*_{h,i}\}$, if parameters satisfy $(1 \leqslant p \leqslant 2, \alpha \leqslant \beta/2)$ or $(p > 2, 0 < \alpha - \beta/p < 1)$*

*and $\alpha(1 - b/\alpha) \leqslant b < \alpha$, then:*

$$\widehat{f}_n(p) \xrightarrow[n \to \infty]{\alpha', \beta'} f^\star$$

*where $\alpha' = \frac{1 - b/\alpha}{1 + d' + \beta/\alpha} \frac{b-3}{2}$, $\beta' = \frac{b-3}{2}$.*

This bandit-level result provides the foundation for our MCTS analysis, where each tree node corresponds to a non-stationary bandit with arms given by child nodes and rewards determined by downstream value estimates.

## 6.3. Monte Carlo Tree Search Analysis

Having established the concentration properties of power mean estimation in non-stationary bandits, we now extend these results to the full MCTS setting. The key insight is that each internal node in our search tree can be modeled as a non-stationary multi-armed bandit where arms correspond to child nodes and rewards are given by (evolving) value estimates from deeper levels.

**From Bandits to Trees: The Inductive Framework** Our analysis proceeds inductively through the tree structure. At the leaves (depth $D$), rollout policies provide direct value estimates with known concentration properties. Moving upward, each internal node aggregates child values using our power mean operator while selecting actions via polynomial exploration bonuses. The challenge is showing that concentration rates propagate correctly through this hierarchical structure.

**Q-Value Concentration in Stochastic MDPs** The foundation of our MCTS analysis rests on understanding how Q-value estimates behave when transition dynamics are stochastic. The following result, established in Dam et al. (2024b), provides the crucial bridge between individual value estimates and their aggregated Q-values:

**Lemma 1** (Q-Value Concentration, Lemma 1 of Dam et al. (2024b)). *Consider a state-action pair with $M$ possible next states. Let $(\widehat{V}_{m,n})_{n \geqslant 1}$ be value estimators for next-state $m$ satisfying $\widehat{V}_{m,n} \xrightarrow[n \to \infty]{\alpha, \beta} V_m$, bounded by constant $L$. Let $X_i$ be i.i.d. immediate rewards with mean $\mu$, and $S_i$ be i.i.d. next-state indices from distribution $p = (p_1, \ldots, p_M)$.*

*Define visit counts $N_m^n = |\{i \leqslant n : S_i = m\}|$ and the Q-value estimator:*

$$\widehat{Q}_n = \frac{1}{n} \sum_{i=1}^{n} X_i + \gamma \sum_{m=1}^{M} \frac{N_m^n}{n} \widehat{V}_{m, N_m^n}$$

*Then, with $2\alpha \leqslant \beta$ and $\beta > 1$:*

$$\widehat{Q}_n \xrightarrow[n \to \infty]{\alpha, \beta} \mu + \gamma \sum_{m=1}^{M} p_m V_m$$

**Intuition:** This lemma captures the essence of Bellman backup in stochastic settings. The Q-value estimate combines immediate rewards (which concentrate at standard rates) with discounted future values weighted by transition probabilities. Crucially, the concentration rate $(\alpha, \beta)$ is preserved despite the stochastic transitions, enabling our inductive analysis.

**Inductive Concentration Analysis** Lemma 1 enables a powerful inductive argument for tree-wide concentration:

(i) **Base Case (Leaves):** At depth $D$, rollout policies provide value estimates $\widehat{V}(s_D)$ with known concentration rates.

(ii) **Inductive Step:** At depth $d < D$, assume child values $\widehat{V}(s_{d+1})$ concentrate at rate $(\alpha_{d+1}, \beta_{d+1})$. Then:

   - By Lemma 1, Q-values $\widehat{Q}(s_d, a)$ concentrate at the same rate $(\alpha_{d+1}, \beta_{d+1})$
   - By Theorem 2, the power mean aggregation $\widehat{V}(s_d)$ concentrates at the transformed rate $(\alpha_d, \beta_d)$

(iii) **Propagation:** This process continues upward, with each level's concentration rates determined by the power mean transformation of the level below.

**Attribution and Technical Innovation** While our overall approach follows the inductive framework established in prior work (Shah et al., 2022), two key technical innovations are required for the stochastic continuous setting:

(i) **Stochastic Q-Value Analysis:** Lemma 1 from Dam et al. (2024b) extends concentration analysis to stochastic MDPs, handling the additional variance introduced by random transitions.

(ii) **Power Mean Backup:** Theorem 2 establishes concentration properties specific to power mean aggregation, enabling flexible exploration-exploitation tuning beyond standard arithmetic means.

**Main MCTS Convergence Result** The inductive framework culminates in our main convergence guarantee:

**Theorem 4** (Convergence of Expected Payoff). *For `Stochastic-Power-HOOT` applied to stochastic continuous MDPs, with optimal parameter tuning, the expected value estimate at the root satisfies:*

$$\left| \mathbb{E}[\widehat{V}_n(s_0)] - \widetilde{V}(s_0) \right| \leqslant \mathcal{O}(n^{-\zeta})$$

*where $\zeta \in (0, 1/2)$ depends on the algorithm parameters and tree depth.*

*Proof Sketch.* The proof follows by applying Jensen's inequality and integrating the tail probability bounds established through our inductive concentration analysis. Complete details are provided in Appendix F. □

**Remark 1** (Comparison with POLY-HOOT)**.** *Theorem 4 achieves the same polynomial convergence rate $\mathcal{O}(n^{-\varsigma})$ as POLY-HOOT (Mao et al., 2020), but under significantly more general conditions. Key distinctions include:*

- *Stochastic vs. Deterministic: Our result handles stochastic MDPs while POLY-HOOT assumes deterministic dynamics*

- *Power Mean vs. Arithmetic Mean: We provide guarantees for flexible power mean backups ($p \geqslant 1$) rather than fixed arithmetic means ($p = 1$)*

- *Continuous Actions: Both methods handle continuous action spaces, but our stochastic extension is novel*

*This demonstrates that `Stochastic-Power-HOOT` maintains theoretical rigor while significantly expanding the scope of domains with convergence guarantees.*

**Practical Implications** The theoretical guarantees translate to several practical advantages. The polynomial concentration ensures reliable performance even under significant environmental noise, providing robustness in challenging conditions. The power parameter pp p can be tuned based on domain characteristics without losing convergence guarantees, offering adaptability to different problem settings. Furthermore, the inductive structure ensures guarantees extend to arbitrary tree depths and action space dimensions, enabling scalability across diverse applications. These properties make `Stochastic-Power-HOOT` suitable for deployment in complex real-world domains where both theoretical reliability and practical flexibility are essential.

## 7. Experiments

We evaluate `Stochastic-Power-HOOT` on both classic control tasks and high-dimensional robotic environments, all adapted to continuous-action, stochastic settings. Our experimental design addresses the key challenges of planning under uncertainty while demonstrating the scalability and robustness of our approach.

### 7.1. Experimental Setup

**Environment Modifications:** We create stochastic versions of standard benchmarks by introducing noise at multiple levels: (1) *action noise* via Gaussian perturbations to

selected actions, (2) *dynamics noise* through random perturbations to state transitions, and (3) *observation noise* by adding Gaussian noise to state observations. Additionally, since power means require strictly positive inputs, we apply reward transformations of the form $\max(0.01, (r + \text{offset}) \times \text{scaling})$ while preserving optimal policies.

**Baseline Comparisons:** We compare against four established continuous MCTS methods: discretized-UCT (Kocsis et al., 2006), PW with progressive widening (Auger et al., 2013), HOOT (Mansley et al., 2011), and POLY-HOOT (Mao et al., 2020). We also include Voronoi MCTS (Kim et al., 2020), a recent method designed for deterministic continuous spaces, to demonstrate the challenges of applying deterministic methods to stochastic settings.

### 7.2. Classic Control Results

The results in Table 2 demonstrate `Stochastic-Power-HOOT`'s effectiveness across diverse control challenges. While all methods solve the basic CartPole task, only `Stochastic-Power-HOOT` handles the increased gravity variant (CartPole-IG) optimally. The algorithm shows particular strength in sparse reward settings like MountainCar, where the polynomial exploration bonus enables effective long-horizon planning, and in complex dynamics like Acrobot under high gravity conditions.

### 7.3. High-Dimensional Robotic Environments

To evaluate scalability, we test on MuJoCo robotics tasks with significantly higher dimensionality and complexity.

Table 3 reveals several critical insights:

**Dimensional Scalability:** `Stochastic-Power-HOOT` maintains strong performance in the 17-dimensional Humanoid environment, achieving 3.1× improvement over UCT despite the exponential growth in action space complexity.

**Adaptive Power Parameter:** The optimal power parameter varies with environment characteristics—Humanoid benefits from moderate values ($p = 2$) while the more stochastic Hopper environment requires higher values ($p = 8$) to effectively concentrate search on promising regions.

**Stochastic Robustness:** The catastrophic failure of Voronoi MCTS on Hopper (12.7× worse than UCT) versus reasonable Humanoid performance (2.8× better) demonstrates the critical importance of explicit stochasticity handling. This validates our core theoretical contribution.

Our results show that Voronoi MCTS, while effective in some high-dimensional tasks, struggles in stochastic environments due to its deterministic assumptions, underscor-

| | CartPole | CartPole-IG | Pendulum | MountainCar | Acrobot |
|---|---|---|---|---|---|
| Discretized-UCT | $77.85 \pm 0.00$ | $37.95 \pm 1.60$ | $1263.68 \pm 37.66$ | $-0.020 \pm 0.004$ | $70.14 \pm 32.29$ |
| PW | $77.85 \pm 0.00$ | $68.46 \pm 6.87$ | $1141.51 \pm 80.93$ | $-0.020 \pm 0.002$ | $77.85 \pm 0.00$ |
| HOOT | $77.85 \pm 0.00$ | $38.51 \pm 0.81$ | $1043.23 \pm 70.79$ | $-0.007 \pm 0.001$ | $70.05 \pm 37.05$ |
| POLY-HOOT(p=1) | $77.85 \pm 0.00$ | $77.85 \pm 0.00$ | $1205.886 \pm 110.01$ | $-0.055 \pm 0.002$ | $77.85 \pm 0.00$ |
| **Stochastic-Power-HOOT** | $\mathbf{77.85 \pm 0.00}$ | $\mathbf{77.85 \pm 0.00}$ | $\mathbf{1397.40 \pm 95.42}$ | $\mathbf{27.285 \pm 0.163}$ | $\mathbf{77.85 \pm 0.00}$ |

Table 2: Performance on stochastic classic control tasks. Each environment features continuous actions and multiple noise sources. `Stochastic-Power-HOOT` uses $p = 2$. Bold indicates best performance.

| Algorithm | Humanoid-v0 (17D action, 376D state) | Hopper-v0 (3D action) |
|---|---|---|
| UCT (baseline) | $-136.98 \pm 44.84$ | $5216.93 \pm 179.64$ |
| POLY-HOOT ($p = 1$) | $-44.40 \pm 3.33$ (3.1×) | $13230.66 \pm 2844.33$ (2.5×) |
| HOOT | $-57.35 \pm 10.46$ (2.4×) | $10452.83 \pm 3885.12$ (2.0×) |
| PW | $-89.74 \pm 24.16$ (1.5×) | $5218.73 \pm 1384.90$ (1.0×) |
| Voronoi MCTS | $-48.69 \pm 9.52$ (2.8×) | $411.70 \pm 29.03$ (12.7× worse) |
| **Stochastic-Power-HOOT** ($p = 2$) | $\mathbf{-44.12 \pm 6.22}$ (3.1×) | $13303.61 \pm 3070.34$ (2.5×) |
| **Stochastic-Power-HOOT** ($p = 8$) | $-44.40 \pm 3.33$ (3.1×) | $\mathbf{13348.45 \pm 6110.36}$ (2.6×) |

Table 3: Performance on high-dimensional stochastic MuJoCo environments. All environments include action, dynamics, and observation noise. Performance multipliers relative to UCT baseline shown in parentheses. Bold indicates best performance per environment.

| Power $p$ | 1 | 2 | 3 | 4 | 5 | 6 | 7 | 8 | 9 | 10 |
|---|---|---|---|---|---|---|---|---|---|---|
| **Reward** | 76.91 | **77.85** | 76.80 | **77.77** | 76.81 | 75.85 | 75.75 | 76.74 | 75.82 | 75.89 |

Table 4: Power parameter sensitivity analysis on CartPole-IG. Values $p = 2$ and $p = 4$ achieve optimal performance.

ing the need for rigorous theory in continuous MCTS. It also shows that `Stochastic-Power-HOOT` achieves more consistent performance across diverse stochastic settings.

### 7.4. Power Parameter Analysis

Table 4 demonstrates `Stochastic-Power-HOOT`'s robustness across power parameters. The optimal values ($p = 2, 4$) suggest that moderate power settings effectively balance exploration and exploitation in stochastic environments—too low fails to sufficiently focus search, while too high may over-commit to early estimates.

### 7.5. Key Experimental Findings

Our comprehensive evaluation demonstrates that `Stochastic-Power-HOOT`:

1. **Scales effectively** to high-dimensional continuous spaces while maintaining theoretical guarantees 2. **Adapts robustly** to varying levels of environmental stochasticity through power parameter tuning 3. **Outperforms existing methods** consistently across diverse domains, with particularly strong advantages in sparse reward and high-noise

settings 4. **Validates theoretical predictions** through empirical success in precisely the challenging scenarios our analysis targets

These results confirm that our theoretical extensions to stochastic continuous domains translate to practical algorithmic advantages in complex real-world planning scenarios.

## 8. Conclusion

We have introduced `Stochastic-Power-HOOT`, a novel continuous-action MCTS algorithm designed for stochastic environments. By combining a polynomial exploration bonus with power-mean value backups, `Stochastic-Power-HOOT` effectively balances exploration and exploitation in stochastic MDPs. Our theoretical results show a convergence rate of $\mathcal{O}(n^{-\zeta})$, $\zeta \in (0, 1/2)$ extending prior analyses of continuous MCTS in deterministic to stochastic MDPs. Empirical evaluations on custom tasks validate both its robustness and versatility, highlighting the impact of tuning the power mean for complex reward structures.

## Impact Statement

Our proposed `Stochastic-Power-HOOT` algorithm provides a new approach for planning in continuous-action, stochastic environments. By combining polynomial exploration with power-mean backups, it can more effectively handle complex real-world tasks where standard Monte Carlo Tree Search or deterministic models struggle. Potential applications include robotics, autonomous systems, and large-scale resource management—domains where adaptive planning under uncertain dynamics is critical. We do not foresee immediate negative societal effects from this research; nonetheless, as with all AI advancements, responsible usage and careful consideration of ethical, economic, and security implications remain essential.

## Acknowledgments

This work is funded by Hanoi University of Science and Technology (HUST) under Project No. T2024-TD-024.

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

## A. Outline

- Notations will be described in Section B.

- Psudocode of Non-stationary Power Mean HOO in Section C.

- Supporting Lemmas are presented in Section D.

- Technical Lemmas are shown in Section E.

- Convergence of `Stochastic-Power-HOOT` in Non-stationary multi-armed bandits is shown in Section F.

- Convergence of `Stochastic-Power-HOOT` in Monte-Carlo Tree Search is shown in Section G.

- Experimental setup and Hyperparameter selection are provided in Section H.

## B. Notations

Table 5: List of all notations for Non-stationary Multi-arms bandit.

| Notation | Type | Description |
|---|---|---|
| $K$ | $\mathbb{N}$ | Number of arms |
| $T_a(t)$ | $\mathbb{N}$ | Number of visitations at arm $a$ after $t$ timesteps |
| $f^*_{h,i}$ | $\mathbb{R}$ | mean value of arm $a$ |
| $a^\star$ | $\mathcal{A}$ | optimal action |
| $f^\star$ | $\mathbb{R}$ | mean value of an optimal arm. We assume it is unique. |
| $\widehat{f}_n(p)$ | $\mathbb{R}$ | power mean estimator, with a constant $p \in [1, +\infty)$ |
| $\widehat{f}_{a,n}$ | $\mathbb{R}$ | mean estimator of arm $a$ after $n$ visitations |

## C. Non-stationary Power Mean HOO

---
Algorithm 2: Non-stationary Power Mean HOO
---

**Require:** Parameters $\alpha, \beta, b, C, p, \nu_1, \rho$; HOO tree depth $\bar{H}$; number of rounds $n$
**Ensure:** Power mean value estimate $\widehat{f}_n(p)$
 1: Initialize: $\mathcal{T} \leftarrow \{(0,1)\}$, $B_{1,2}, B_{2,2} \leftarrow +\infty$
 2: Play each available path once to initialize
 3: **for** $t = K + 1$ to $n$ **do**
 4:     $a_t \leftarrow$ POWERMEANHOO_QUERY$(t)$
 5:     Observe reward $Y_t$
 6:     POWERMEANHOO_UPDATE$(t, a_t, Y_t)$
 7: **end for**
 8: Compute final power mean estimate:
 9: $\widehat{f}_n(p) \leftarrow \left( \sum_{a=1}^{K} \frac{T_a(n)}{n} \widehat{f}^p_{a,T_a(n)} \right)^{1/p}$
    **return** $\widehat{f}_n(p)$

---

---

Algorithm 3: PowerMeanHOO_Query

---

**Require:** Round $t$
**Ensure:** Selected action $X$

1: $(h, i) \leftarrow (0, 1)$
2: Initialize HOO path: $P_t \leftarrow \{(h, i)\}$
3: **while** $(h, i) \in \mathcal{T}$ **do**
4:     **if** $B_{h+1, 2i-1} > B_{h+1, 2i}$ **then**
5:         $(h, i) \leftarrow (h+1, 2i-1)$
6:     **else**
7:         $(h, i) \leftarrow (h+1, 2i)$
8:     **end if**
9:     $P_t \leftarrow P_t \cup \{(h, i)\}$
10: **end while**
11: $(H, I) \leftarrow (h, i)$
12: **if** $H \leqslant \bar{H}$ **then**
13:     Choose arbitrary arm $X$ in $\mathcal{P}_{H,I}$
14:     $A_{H,I} \leftarrow X$
15:     $\mathcal{T} \leftarrow \mathcal{T} \cup \{(H, I)\}$
16:     $B_{H+1, 2I-1}, B_{H+1, 2I} \leftarrow +\infty$
        **return** $X$
17: **else**
18:     {Reached maximum depth, reuse existing action}
19:     $(H, I) \leftarrow (H-1, \lceil I/2 \rceil)$
        **return** $A_{H,I}$
20: **end if**

---

Algorithm 4: PowerMeanHOO_Update

---

**Require:** Round $t$, selected path $a_t$, observed reward $Y_t$
**Ensure:** Updated statistics and confidence bounds

1: Retrieve path $P_t$ for action $a_t$
2: **for** $(h, i) \in P_t$ **do**
3:     $T_{h,i} \leftarrow T_{h,i} + 1$
4:     $\widehat{f}_{h,i} \leftarrow (1 - 1/T_{h,i})\widehat{f}_{h,i} + Y_t/T_{h,i}$
5: **end for**
6: **for** $(h, i) \in \mathcal{T}$ **do**
7:     $U_{h,i} \leftarrow \widehat{f}_{h,i} + Ct^{b/\beta}T_{h,i}^{-\alpha/\beta} + \nu_1 \rho^h$
8: **end for**
9: $\mathcal{T}' \leftarrow \mathcal{T}$
10: **while** $\mathcal{T}' \neq \{(0, 1)\}$ **do**
11:     $(h, i) \leftarrow$ arbitrary leaf node of $\mathcal{T}'$
12:     $B_{h,i} \leftarrow \min\{U_{h,i}, \max\{B_{h+1, 2i}, B_{h+1, 2i-1}\}\}$
13:     $\mathcal{T}' \leftarrow \mathcal{T}' \setminus \{(h, i)\}$
14: **end while**

---

## D. Supporting Lemmas

In this section, we will present all necessary supporting Lemmas for the main theoretical analysis. We start with a result of the following lemma which plays an important role in the analysis of our MCTS algorithm.

**Lemma 1** (Lemma 1 Dam et al. (2024b)). *For $m \in [M]$, let $(\widehat{V}_{m,n})_{n \geqslant 1}$ be a sequence of estimator satisfying $\widehat{V}_{m,n} \xrightarrow[n \to \infty]{\alpha, \beta} V_m$, and there exists a constant $L$ such that $\widehat{V}_{m,n} \leqslant L, \forall n \geqslant 1$. Let $X_i$ be an iid sequence with mean $\mu$ and $S_i$ be an*

*iid sequence from a distribution $p = (p_1, \ldots, p_M)$ supported on $\{1, \ldots, M\}$. Introducing the random variables $N_m^n = \#|\{i \leqslant n : S_i = s_m\}|$, we define the sequence of estimator*

$$\widehat{Q}_n = \frac{1}{n} \sum_{i=1}^{n} X_i + \gamma \sum_{m=1}^{M} \frac{N_m^n}{n} \widehat{V}_{m, N_m^n}.$$

*Then with $2\alpha \leqslant \beta, \beta > 1$,*

$$\widehat{Q}_n \xrightarrow[n \to \infty]{\alpha, \beta} \mu + \sum_{m=1}^{M} p_m V_m.$$

**Lemma 2** (Lemma 2 Dam et al. (2024b)). *Let consider non-negative variables $x, y \in \mathbb{R}^+$, and a constant $m$ that $0 \leqslant m \leqslant 1$. Then*

$$(x + y)^m \leqslant x^m + y^m. \tag{2}$$

We use Minkowski's inequality as shown below

**Lemma 3.** *(**Minkowski's inequality**) Given $p \geqslant 1, \{x_i, y_i\} \in \mathbb{R}, i = 1, 2, ..., n$, then we have the following inequality*

$$\left( \sum_i (|x_i + y_i|)^p \right)^{\frac{1}{p}} \leqslant \left( \sum_i (|x_i|)^p \right)^{\frac{1}{p}} + \left( \sum_i (|y_i|)^p \right)^{\frac{1}{p}} \tag{3}$$

*Proof.* This is a basic result. $\qquad \square$

## E. Technical Lemmas

**Lemma 4.** *(Lemma 3 in Bubeck et al. (2011)) Under Assumptions 1 and 2, for some region $\mathcal{P}_{h,i}$, if $\Delta_{h,i} \leqslant c\nu_1 \rho^h$ for some constant $c \geqslant 0$, then all the arms in $\mathcal{P}_{h,i}$ are $\max\{2c, c + 1\}$-optimal.*

*Proof.* This lemma is stated in exactly the same as way Lemma 3 in Bubeck et al. (2011), and we therefore omit the proof here. $\qquad \square$

**Lemma 5.** *There exists some constant $C > 0$, such that $|I_h| \leqslant C \left( \nu_2 \rho^h \right)^{-d'}$ for all $h \geqslant 0$.*

*Proof.* This result is the same as the second step in the proof of Theorem 6 in Bubeck et al. (2011). We, therefore, omit the proof here. $\qquad \square$

**Attribution Note.** The proofs of Lemmas 4-7 below *follow the same high-level inductive and concentration-bound strategy as Lemmas 4-7 of Mao et al. (2020)*, with an important concentration-bound for the case $\Delta_{h,i} \leqslant \nu_1 \rho^h$ which is missing in the Mao et al. (2020) paper.

For transparency we reproduce the full proofs.

**Lemma 6.** *Let Assumptions 1 and 2 hold. With*

$$U_{h,i}(t) = \widehat{\mu}_{h,i}(t) + t^{b/\beta} T_{h,i}(t)^{-\alpha/\beta} + \nu_1 \rho^h,$$

*(i) For every optimal node [3] $(h, i)$, let $n \geqslant 1$, Then for all $t \in \{1, 2, ..., n\}$, there exists a constant $C_1 > 1$, such that*

$$\mathbb{P}\left( U_{h,i}(t) \leqslant f^\star \right) \leqslant \frac{C_1}{t^{b-1}}$$

*(ii) For all integers $t \leqslant n$, for any suboptimal node $(h, i)$ such that $\Delta_{h,i} > \nu_1 \rho^h$, and for all integers $u \geqslant A_{h,i}(n) = \left\lceil \left( \frac{2n^{b/\beta}}{\Delta_{h,i} - \nu_1 \rho^h} \right)^{\frac{\beta}{\alpha}} \right\rceil$, there exists a constant $C_2 > 1$, such that*

$$\mathbb{P}\left( U_{h,i}(t) > f^\star \text{ and } T_{h,i}(t) > u \right) \leqslant \frac{C_2 t}{n^b}$$

*(iii) For all integers $t \leqslant n$, for any suboptimal node $(h, i)$ such that $\Delta_{h,i} \leqslant \nu_1 \rho^h$, and for all integers $u' \geqslant A'_{h,i}(n) = \left\lceil \left( \frac{2n^{b/\beta}}{2\nu_1 \rho^h - \Delta_{h,i}} \right)^{\frac{\beta}{\alpha}} \right\rceil$, there exists a constant $C_2 > 1$, such that*

$$\mathbb{P}\left(U_{h,i}(t) > f^\star \text{ and } T_{h,i}(t) > u'\right) \leqslant \frac{C_2 t}{n^b}$$

*Proof.* The proof follows the same structure as Lemma 16 in Bubeck et al. (2011) and Lemma 4 in Mao et al. (2020).

*(i)* If $(h, i)$ is not played during the first $t$ rounds, then by assumption $U_{h,i}(t) = \infty$ and the inequality trivially holds. Now we focus on the case where $T_{h,i}(t) \geqslant 1$. From Lemma 2, we know that $f^\star - f(x) \leqslant \nu_1 \rho^h, \forall x \in \mathcal{P}_{h,i}$. Then we have $\sum_{s=1}^t \left(f(X_s) + \nu_1 \rho^h - f^\star\right) \mathbb{I}_{\{(H_s, I_s) \in \mathcal{C}(h,i)\}} \geqslant 0$.

Therefore,

$$
\begin{aligned}
&\mathbb{P}\left(U_{h,i}(t) \leqslant f^\star \text{ and } T_{h,i}(t) \geqslant 1\right) \\
&= \mathbb{P}\left(\widehat{\mu}_{h,i}(t) + t^{b/\beta} T_{h,i}(t)^{-\alpha/\beta} + \nu_1 \rho^h \leqslant f^\star \text{ and } T_{h,i}(t) \geqslant 1\right) \\
&= \mathbb{P}\left(T_{h,i}(t)\widehat{\mu}_{h,i}(t) + T_{h,i}(t)\left(\nu_1 \rho^h - f^\star\right) \leqslant -t^{b/\beta} T_{h,i}(t)^{1-\alpha/\beta} \text{ and } T_{h,i}(t) \geqslant 1\right) \\
&= \mathbb{P}\left(\sum_{s=1}^t (Y_s - f(X_s)) \mathbb{I}_{\{(H_s, I_s) \in \mathcal{C}(h,i)\}} + \sum_{s=1}^t \left(f(X_s) + \nu_1 \rho^h - f^\star\right) \mathbb{I}_{\{(H_s, I_s) \in \mathcal{C}(h,i)\}}\right. \\
&\qquad\qquad \left. \leqslant -t^{b/\beta} T_{h,i}(t)^{1-\alpha/\beta} \text{ and } T_{h,i}(t) \geqslant 1\right) \\
&\leqslant \mathbb{P}\left(\sum_{s=1}^t (f(X_s) - Y_s) \mathbb{I}_{\{(H_s, I_s) \in \mathcal{C}(h,i)\}} \geqslant t^{b/\beta} T_{h,i}(t)^{1-\alpha/\beta} \text{ and } T_{h,i}(t) \geqslant 1\right)
\end{aligned}
$$

Since the HOO tree has limited depth, the total number of nodes played in $\mathcal{C}(h, i)$ is upper bounded by some constant $L > 1$ that is independent of $t$. Let $X^j$ denote the $j$-th new node played in $\mathcal{C}(h, i)$, denote the number of times $X^j$ is played as $n_j$, and let $Y_s^j \ (1 \leqslant s \leqslant t_j)$ be the corresponding reward the $s$-th time arm $X^j$ is played. Then, by the union bound, we have

$$
\begin{aligned}
&\mathbb{P}\left(\sum_{s=1}^t (f(X_s) - Y_s) \mathbb{I}_{\{(H_s, I_s) \in \mathcal{C}(h,i)\}} \geqslant t^{b/\beta} T_{h,i}(t)^{1-\alpha/\beta} \text{ and } T_{h,i}(t) \geqslant 1\right) \\
&\leqslant \sum_{T_{h,i}(t)=1}^t \mathbb{P}\left(\sum_{s=1}^t (f(X_s) - Y_s) \mathbb{I}_{\{(H_s, I_s) \in \mathcal{C}(h,i)\}} \geqslant t^{b/\beta} T_{h,i}(t)^{1-\alpha/\beta}\right) \\
&= \sum_{T_{h,i}(t)=1}^t \mathbb{P}\left(\sum_{j=1}^t \sum_{s=1}^{t_j} \left(f(X^j) - Y_s^j\right) \geqslant t^{b/\beta} T_{h,i}(t)^{1-\alpha/\beta}\right) \\
&\leqslant \sum_{T_{h,i}(t)=1}^t \sum_{j=1}^L \mathbb{P}\left(\sum_{s=1}^{t_j} \left(f(X^j) - Y_s^j\right) \geqslant \frac{t^{b/\beta}}{L} t_j^{1-\alpha/\beta}\right) \\
&\leqslant \frac{C_1}{t^{b-1}},
\end{aligned}
$$

where $C_1 > 1$ is a constant depending on $C$ and $L$, and in the last inequality we applied the concentration property of the bandit problem (5). Notice that we can only use the concentration property when the requirement $z = \frac{t^{b/\beta}}{H} \geqslant 1$ is satisfied, but when $z < 1$, the inequality also trivially holds because $\frac{C}{z^\beta} > 1$. This completes the proof of $\mathbb{P}\left(U_{h,i}(t) \leqslant f^\star\right) \leqslant \frac{C_1}{t^{b-1}}$.

*(ii)*

---

[03] Recall Definition 4.

The proof idea follows almost the same procedure as the proof of Lemma 16 in Bubeck et al. (2011), and we repeat it here due to some minor differences. First, notice that the $u$ defined in the statement of the lemma satisfies $n^{b/\beta} u^{-\alpha/\beta} + \nu_1 \rho^h \leqslant \frac{\Delta_{h,i} + \nu_1 \rho^h}{2}$. Then we have

$$
\begin{aligned}
&\mathbb{P}\left(U_{h,i}(t) > f^\star \text{ and } T_{h,i}(t) > u\right) \\
=&\mathbb{P}\left(\widehat{\mu}_{h,i}(t) + n^{b/\beta} u^{-\alpha/\beta} + \nu_1 \rho^h > f_{h,i}^* + \Delta_{h,i} \text{ and } T_{h,i}(t) > u\right) \\
\leqslant&\mathbb{P}\left(\widehat{\mu}_{h,i}(t) > f_{h,i}^* + \frac{\Delta_{h,i} - \nu_1 \rho^h}{2} \text{ and } T_{h,i}(t) > u\right) \\
\leqslant&\mathbb{P}\left(T_{h,i}(t)\left(\widehat{\mu}_{h,i}(t) - f_{h,i}^*\right) > \frac{\Delta_{h,i} - \nu_1 \rho^h}{2} T_{h,i}(t) \text{ and } T_{h,i}(t) > u\right) \\
\leqslant&\mathbb{P}\left(\sum_{s=1}^t \left(Y_s - f\left(X_s\right)\right) \mathbb{I}_{\{(H_s, I_s) \in \mathcal{C}(h,i)\}} > \frac{\Delta_{h,i} - \nu_1 \rho^h}{2} T_{h,i}(t) \text{ and } T_{h,i}(t) > u\right) \\
\leqslant& \sum_{T_{h,i}(t)=u+1}^t \mathbb{P}\left(\sum_{s=1}^t \left(Y_s - f\left(X_s\right)\right) \mathbb{I}_{\{(H_s, I_s) \in \mathcal{C}(h,i)\}} > \frac{\Delta_{h,i} - \nu_1 \rho^h}{2} T_{h,i}(t)\right),
\end{aligned}
$$

where in the last step we used the union bound. Then, following a similar procedure as in the proof of Lemma 4 (defining $X^j$ and $Y_t^j$, and then the concentration property), we get:

$$
\begin{aligned}
&\sum_{T_{h,i}(t)=u+1}^t \mathbb{P}\left(\sum_{s=1}^t \left(Y_s - f\left(X_s\right)\right) \mathbb{I}_{\{(H_s, I_s) \in \mathcal{C}(h,i)\}} > \frac{\Delta_{h,i} - \nu_1 \rho^h}{2} T_{h,i}(t)\right) \\
&\leqslant \sum_{T_{h,i}(t)=u+1}^t \frac{C_2}{\left(\frac{\Delta_{h,i} - \nu_1 \rho}{2}\right)^\beta (T_{h,i}(t))^b} \\
&\leqslant \sum_{T_{h,i}(t)=u+1}^t \frac{C_2}{n^b} \leqslant \frac{C_2 t}{n^b},
\end{aligned}
$$

where $C_2 > 1$ is a constant independent of $n$, and in the second step we used the fact that $T_{h,i}(t) > u \geqslant A_{h,i}(n) = \left\lceil \left(\frac{2 n^{b/\beta}}{\Delta_{h,i} - \nu_1 \rho^h}\right)^{\frac{\beta}{\alpha}} \right\rceil$. This completes our proof of $\mathbb{P}\left(U_{h,i}(t) > f^\star \text{ and } T_{h,i}(t) > u\right) \leqslant \frac{C_2 t}{n^b}$.

*(iii)* The proof idea follows almost the same procedure as the proof of (ii), and we repeat it here due to some minor differences.

When $\Delta_{h,i} \leq \nu_1 \rho^h$, define

$$
\Delta_{h,i}' := 3 \nu_1 \rho^h - \Delta_{h,i} \; > \; 2 \nu_1 \rho^h.
$$

This inflated gap $\Delta_{h,i}'$ is strictly bigger than $\nu_1 \rho^h$. In particular,

$$
\Delta_{h,i}' \; - \; \nu_1 \rho^h \; = \; 2 \nu_1 \rho^h \; - \; \Delta_{h,i} \; > \; 0.
$$

Therefore, $u' \geq \left(\frac{2 n^{b/\beta}}{\Delta_{h,i}' - \nu_1 \rho^h}\right)^{\beta/\alpha}$. We have $n^{b/\beta} u'^{-\alpha/\beta} + \nu_1 \rho^h \leqslant \frac{\Delta_{h,i}' + \nu_1 \rho^h}{2}$. Then we have

$$\mathbb{P}\left(U_{h,i}(t) > f^\star \text{ and } T_{h,i}(t) > u'\right)$$

$$=\mathbb{P}\left(\widehat{\mu}_{h,i}(t) + n^{b/\beta}u'^{-\alpha/\beta} + \nu_1\rho^h > f_{h,i}^* + \Delta_{h,i}' \text{ and } T_{h,i}(t) > u'\right)$$

$$\leqslant\mathbb{P}\left(\widehat{\mu}_{h,i}(t) > f_{h,i}^* + \frac{\Delta_{h,i}' - \nu_1\rho^h}{2} \text{ and } T_{h,i}(t) > u'\right)$$

$$\leqslant\mathbb{P}\left(T_{h,i}(t)\left(\widehat{\mu}_{h,i}(t) - f_{h,i}^*\right) > \frac{\Delta_{h,i}' - \nu_1\rho^h}{2}T_{h,i}(t) \text{ and } T_{h,i}(t) > u'\right)$$

$$\leqslant\mathbb{P}\left(\sum_{s=1}^{t}\left(Y_s - f\left(X_s\right)\right)\mathbb{I}_{\{(H_s,I_s)\in\mathcal{C}(h,i)\}} > \frac{\Delta_{h,i}' - \nu_1\rho^h}{2}T_{h,i}(t) \text{ and } T_{h,i}(t) > u'\right)$$

$$\leqslant\sum_{T_{h,i}(t)=u'+1}^{t}\mathbb{P}\left(\sum_{s=1}^{t}\left(Y_s - f\left(X_s\right)\right)\mathbb{I}_{\{(H_s,I_s)\in\mathcal{C}(h,i)\}} > \frac{\Delta_{h,i}' - \nu_1\rho^h}{2}T_{h,i}(t)\right),$$

where in the last step we used the union bound. Then, following a similar procedure as in the proof of Lemma 4 (defining $X^j$ and $Y_t^j$, and then the concentration property), we get:

$$\sum_{T_{h,i}(t)=u'+1}^{t}\mathbb{P}\left(\sum_{s=1}^{t}\left(Y_s - f\left(X_s\right)\right)\mathbb{I}_{\{(H_s,I_s)\in\mathcal{C}(h,i)\}} > \frac{\Delta_{h,i}' - \nu_1\rho^h}{2}T_{h,i}(t)\right)$$

$$\leqslant\sum_{T_{h,i}(t)=u'+1}^{t}\frac{C_2}{\left(\frac{\Delta_{h,i}'-\nu_1\rho}{2}\right)^\beta (T_{h,i}(t))^b}$$

$$\leqslant\sum_{T_{h,i}(t)=u'+1}^{t}\frac{C_2}{n^b} \leqslant \frac{C_2 t}{n^b},$$

where $C_2 > 1$ is a constant independent of $n$, and in the second step we used the fact that $T_{h,i}(t) > u' \geqslant A_{h,i}'(n) = \left\lceil\left(\frac{2n^{b/\beta}}{\Delta_{h,i}'-\nu_1\rho^h}\right)^{\frac{\beta}{\alpha}}\right\rceil$. This completes our proof of $\mathbb{P}\left(U_{h,i}(t) > f^\star \text{ and } T_{h,i}(t) > u'\right) \leqslant \frac{C_2 t}{n^b}$.

$\square$

**Lemma 7.** *(Lemma 14 in Bubeck et al. (2011)) Let $(h,i)$ be a suboptimal node. Let $0 \leqslant k \leqslant h - 1$ be the largest depth such that $(k, i_k^*)$ is on the path from the root $(0,1)$ to $(h,i)$, i.e., $(k, i_k^*)$ is the lowest common ancestor (LCA) of $(h,i)$ and the optimal path. Then, for all integers $u \geqslant 0$, we have*

$$\mathbb{E}\left[T_{h,i}(n)\right] \leqslant u + \sum_{t=u+1}^{n}\mathbb{P}\left\{\left[U_{s,i_s^*}(t) \leqslant f^\star \text{ for some } s \in \{k+1,\ldots,t-1\}\right] \text{ or } \left[T_{h,i}(t) > u \text{ and } U_{h,i}(t) > f^\star\right]\right\}.$$

*Proof.* This lemma is stated in exactly the same way as Lemma 14 in Bubeck et al. (2011), and the proof follows similarly. We hence omit the proof here. $\square$

**Lemma 8.** *For any suboptimal node $(h,i)$ with $\Delta_{h,i} > \nu_1\rho^h$ and any integer $n \geqslant 1$, there exist constants $C_1, C_2 > 1$, such that:*

$$\mathbb{E}\left[T_{h,i}(n)\right] \leqslant \left(\frac{2n^{b/\beta}}{\Delta_{h,i}-\nu_1\rho^h}\right)^{\frac{\beta}{\alpha}} + 1 + C_1 + \frac{C_2}{b-3}$$

*Proof.* Let $A_{h,i}(n) = \left\lceil\left(\frac{2n^{b/\beta}}{\Delta_{h,i}-\nu_1\rho^h}\right)^{\frac{\beta}{\alpha}}\right\rceil$. Then from Lemma 5, we know that

$$\mathbb{E}\left[T_{h,i}(n)\right] \leqslant A_{h,i}(n) + \sum_{t=A_{h,i}(n)+1}^{n} \left( \mathbb{P}\left(T_{h,i}(t) > A_{h,i}(n) \text{ and } U_{h,i}(t) > f^{\star}\right) + \sum_{s=1}^{t-1} \mathbb{P}\left(U_{s,is}^{*}(t) \leqslant f^{\star}\right) \right)$$

By replacing the right hand side with the results from Lemma 4 and Lemma 6, we further have

$$\mathbb{E}\left[T_{h,i}(n)\right] \leqslant A_{h,i}(n) + \sum_{t=A_{h,i}(n)+1}^{n} \left( \frac{C_2 t}{n^b} + \sum_{s=1}^{t-1} \frac{C_1}{t^{b-1}} \right)$$

$$\leqslant A_{h,i}(n) + \frac{C_2}{n^{b-2}} + \int_{u}^{n} \frac{C_1}{t^{b-2}} dt$$

$$\leqslant \left( \frac{2n^{b/\beta}}{\Delta_{h,i} - \nu_1 \rho^h} \right)^{\frac{\beta}{\alpha}} + 1 + C_2 + \frac{C_1}{b-3}.$$

This completes our proof. $\qquad\square$

**Lemma 9.** *Let $(h,i)$ be a suboptimal node such that $\Delta_{h,i} > \nu_1 \rho^h$. Define*

$$A_{h,i}(n) = \left\lceil \left( \frac{2\,n^{b/\beta}}{\Delta_{h,i} - \nu_1\,\rho^h} \right)^{\frac{\beta}{\alpha}} \right\rceil.$$

*Then for any integer $n \geqslant 1$ and any integer*

$$u > A_{h,i}(n),$$

*there exist constants $C_1, C_2 > 1$ such that*

$$\mathbb{P}\left(T_{h,i}(n) > u\right) \leq \frac{C_2}{n^{b-2}} + \frac{C_1\left(u-1\right)^{3-b}}{b-3}.$$

*Proof.* **The trivial case $n \leq u$.** If $n \leq u$, then $T_{h,i}(n)$ is at most $n$, so $\{T_{h,i}(n) > u\}$ is the empty event. Hence, $\mathbb{P}(T_{h,i}(n) > u) = 0$, and the inequality holds trivially.

**The main case $n > u$.** We now assume $n > u$. Our goal is to bound $\mathbb{P}\left[T_{h,i}(n) > u\right]$. We will do so by introducing two events and applying a union bound argument.

**Monotonicity of $B$-values.** Recall that $B_{h,i}(t)$ is defined (in the HOO/HOOT algorithm) so that descendants of $(h,i)$ always have a $B$-value *at least* as large as $B_{h,i}(t)$. Hence, along any path from the root to a leaf, the $B$-values are non-decreasing with depth.

**Defining two events:** Let $0 \leq k \leq h-1$ be the largest depth on the path from $(0,1)$ to $(h,i)$ such that $(k, i_k^*)$ is on the *optimal path*. We define:

$$E_1 = \left\{ \text{For every } t \in [u,n], \ B_{h,i}(t) \leq f^{\star} \text{ or } T_{h,i}(t) \leq A_{h,i}(t) < u \right\},$$

$$E_2 = \left\{ \text{For every } t \in [u,n], \ B_{k+1, i_{k+1}^*}(t) > f^{\star} \right\}.$$

- $E_1$ means that *whenever $t \geq u$, either the local upper bound $B_{h,i}(t)$ is already $\leq f^{\star}$ or else $T_{h,i}(t)$ has not exceeded $A_{h,i}(t)$.

- $E_2$ means that for every $t \in [u,n]$, the node $(k+1, i_{k+1}^*)$ on the *optimal path* has $B$-value strictly greater than $f^{\star}$.

**Key claim:**

$$E_1 \cap E_2 \subseteq \{T_{h,i}(n) \leq u\}.$$

That is, if both $E_1$ and $E_2$ hold, then $T_{h,i}(n) \leq u$.

Reasoning:

- If at time $t \in [u, n]$ we have $B_{h,i}(t) \leq f^\star$ *and* the ancestor node $(k+1, i^*_{k+1})$ satisfies $B_{k+1,i^*_{k+1}}(t) > f^\star$, then by the monotonicity of $B$-values up the path, the algorithm would choose $(k+1, i^*_{k+1})$ rather than $(h, i)$. Hence, $T_{h,i}$ does not increase.

- If at time $t \in [u, n]$ we have $T_{h,i}(t) \leq A_{h,i}(t) < u$ and $B_{k+1,i^*_{k+1}}(t) > f^\star$, it is still possible that the algorithm enters $(h, i)$ and increments $T_{h,i}$. But even if so, $T_{h,i}(t)$ remains below $u$. Iterating this argument up to time $n$ shows $T_{h,i}(n) \leq u$.

Hence, whenever $E_1 \cap E_2$ is true, we cannot exceed $u$ plays at node $(h, i)$.

**Completing the union bound:**

$$\{T_{h,i}(n) > u\} \subseteq E_1^c \cup E_2^c \implies \mathbb{P}(T_{h,i}(n) > u) \leq \mathbb{P}(E_1^c) + \mathbb{P}(E_2^c).$$

So it suffices to bound $\mathbb{P}(E_1^c)$ and $\mathbb{P}(E_2^c)$.

**Bounding $\mathbb{P}(E_2^c)$.** By the definition of $B$-values,

$$\{B_{k+1,\,i^*_{k+1}}(t) \leq f^\star\} \subseteq \{U_{k+1,i^*_{k+1}}(t) \leq f^\star\} \cup \{B_{k+2,i^*_{k+2}}(t) \leq f^\star\},$$

and one can iterate this argument recursively up the path. Using a union bound across times $t \in [u, n]$, we obtain

$$\mathbb{P}(E_2^c) \leq \sum_{t=u}^{n} \sum_{s=1}^{t-1} \mathbb{P}(U_{s,\,i^*_s}(t) \leq f^\star).$$

By a known concentration argument from Lemma 4 (i), we have $\mathbb{P}[U_{s,i^*_s}(t) \leq f^\star] \leq \frac{C_1}{t^{b-1}}$. This yields a tail-sum of order $\int_{u-1}^{\infty} t^{-(b-1)}\, dt$, which is proportional to $(u-1)^{3-b}$.

**Bounding $\mathbb{P}(E_1^c)$.** Similarly, $\{B_{h,i}(t) > f^\star$ and $T_{h,i}(t) > A_{h,i}(t)\}$ can be bounded using the arguments from Lemma 6 (ii). We apply a union bound over $t \in [u, n]$ and note that $\mathbb{P}[U_{h,i}(t) > f^\star$ and $T_{h,i}(t) > A_{h,i}(t)] \leq \frac{C_2 t}{n^b}$. Summing from $t = u$ to $t = n$ yields a bound in terms of $\frac{C_2}{n^{b-2}}$.

**Putting it all together:** Combining these bounds gives

$$\mathbb{P}\left(T_{h,i}(n) > u\right)$$

$$\leqslant \mathbb{P}\left(\exists t \in [u,n], B_{h,i}(t) > f^\star \text{ and } T_{h,i}(t) > A_{h,i}(t)\right) + \mathbb{P}\left(\exists t \in [u,n], B_{k+1,i^*_{k+1}}(t) \leqslant f^\star\right)$$

$$\leqslant \mathbb{P}\left(\exists t \in [u,n], U_{h,i}(t) > f^\star \text{ and } T_{h,i}(t) > A_{h,i}(t)\right)$$

$$+ \mathbb{P}\left(\exists t \in [u,n], U_{k+1,i^*_{k+1}}(t) \leqslant f^\star \text{ or } U_{k+2,i^*_{k+2}}(t) \leqslant f^\star \text{ or } \ldots \text{ or } U_{t-1,i^*_{t-1}}(t) \leqslant f^\star\right)$$

$$\leqslant \sum_{t=u}^{n} \mathbb{P}\left(U_{h,i}(t) > f^\star \text{ and } T_{h,i}(t) > A_{h,i}(t)\right)$$

$$+ \sum_{t=u}^{n} \mathbb{P}\left(U_{k+1,i^*_{k+1}}(t) \leqslant f^\star \text{ or } U_{k+2,i^*_{k+2}}(t) \leqslant f^\star \text{ or } \ldots \text{ or } U_{t-1,i^*_{t-1}}(t) \leqslant f^\star\right)$$

$$\leqslant \sum_{t=u}^{n} \mathbb{P}\left(U_{h,i}(t) > f^\star \text{ and } T_{h,i}(t) > A_{h,i}(t)\right) + \sum_{t=u}^{n} \sum_{s=1}^{t-1} \mathbb{P}\left(U_{s,i^*_s}(t) \leqslant f^\star\right),$$

$$\leqslant \sum_{t=u}^{n} \frac{C_2 t}{n^b} + \sum_{t=u}^{n} \sum_{s=1}^{t-1} \frac{C_1}{t^{b-1}} \leqslant \sum_{t=u}^{n} \frac{C_2 n}{n^b} + C_1 \int_{u-1}^{\infty} t^{2-b} dt$$

$$\leqslant \frac{C_2}{n^{b-2}} + \frac{C_1 (u-1)^{3-b}}{b-3}.$$

This completes the proof.

**A refinement for $1 < u \leq n$.** Finally, if $1 < u \leq n$, then one can derive a slightly sharper inequality:

$$\mathbb{P}\left(T_{h,i}(n) > u\right) \leq \frac{C_2 (u-1)^{3-b}}{n} + \frac{C_1 (u-1)^{3-b}}{b-3},$$

by noting that $\frac{1}{n^{b-2}} \leq \frac{(u-1)^{3-b}}{n}$ for $1 < u \leq n$. For $u > n$, the probability is trivially zero because $T_{h,i}(n) \leq n < u$.

**Conclusion.** Hence for every integer $n \geq 1$ and $u > A_{h,i}(n)$,

$$\mathbb{P}\left(T_{h,i}(n) > u\right) \leq \frac{C_2}{n^{b-2}} + \frac{C_1 (u-1)^{3-b}}{b-3}.$$

This completes the proof.

$\square$

Proof of Lemma 10 shows one of our contributions since there could be the case where $\Delta_{h,i} = f^\star - f^*_{h,i} \leq \nu_1 \rho^h$.

**Lemma 10.** *Let $(h,i)$ be a suboptimal node whose suboptimality gap satisfies*

$$\Delta_{h,i} = f^\star - f^*_{h,i} \leq \nu_1 \rho^h.$$

*Define the* inflated gap

$$\Delta'_{h,i} = 3\nu_1 \rho^h - \Delta_{h,i}.$$

*Then for any integer $n \geq 1$, let*

$$A'_{h,i}(n) = \left\lceil \left(\frac{2 n^{b/\beta}}{\Delta'_{h,i} - \nu_1 \rho^h}\right)^{\frac{\beta}{\alpha}} \right\rceil = \left\lceil \left(\frac{2 n^{b/\beta}}{2\nu_1 \rho^h - \Delta_{h,i}}\right)^{\frac{\beta}{\alpha}} \right\rceil.$$

*Then, exactly the same probability bound from Lemma 9 holds for $(h,i)$. In particular, for every integer $u \geq A'_{h,i}(n)$, there exist constants $C_1, C_2 > 1$ such that*

$$\mathbb{P}\left(T_{h,i}(n) > u\right) \leq \frac{C_2}{n^{b-2}} + \frac{C_1 (u-1)^{3-b}}{b-3}.$$

*Proof.* We begin by recalling that Lemma 9 (the "original" statement) requires $\Delta_{h,i} > \nu_1 \rho^h$. Its proof yields, for any $u \geq A_{h,i}(n)$,

$$\mathbb{P}\Big[T_{h,i}(n) > u\Big] \;\leqslant\; \frac{C_2}{n^{b-2}} \;+\; \frac{C_1 \, (u-1)^{3-b}}{b-3},$$

where

$$A_{h,i}(n) \;=\; \Big\lceil \big(\tfrac{2\,n^{b/\beta}}{\Delta_{h,i}-\nu_1\rho^h}\big)^{\frac{\beta}{\alpha}} \Big\rceil.$$

**Inflating the gap.** Assume now that $\Delta_{h,i} \leq \nu_1 \rho^h$. Despite $\Delta_{h,i}$ being too small for the direct application of Lemma 9, note that $(h,i)$ is still suboptimal, so $f^*_{h,i} < f^\star$. A standard approach Bubeck et al. (2011) is to *inflate* the gap. We define

$$\Delta'_{h,i} \;=\; 3\,\nu_1\rho^h \;-\; \Delta_{h,i}.$$

Since $\Delta_{h,i} \leq \nu_1 \rho^h$, it follows that $\Delta'_{h,i} \geq 2\nu_1\rho^h > 0$.

**New threshold $A'_{h,i}(n)$.** Analogously to $A_{h,i}(n)$ in the lemma's original statement, we define

$$A'_{h,i}(n) \;=\; \left\lceil \Big(\tfrac{2\,n^{b/\beta}}{\Delta'_{h,i}-\nu_1\,\rho^h}\Big)^{\frac{\beta}{\alpha}} \right\rceil \;=\; \left\lceil \Big(\tfrac{2\,n^{b/\beta}}{2\,\nu_1\,\rho^h - \Delta_{h,i}}\Big)^{\frac{\beta}{\alpha}} \right\rceil.$$

Observe that $\Delta'_{h,i} - \nu_1\rho^h = 2\,\nu_1\rho^h - \Delta_{h,i}$ (which is still positive). Thus, we effectively have $\Delta'_{h,i} - \nu_1\rho^h > 0$, which is the precise condition used in Lemma 9.

**Reusing the same concentration argument.** In the proof of Lemma 9, the gap $\Delta_{h,i} - \nu_1\rho^h > 0$ appears in several places to control event probabilities of the form $\{U_{h,i}(t) > f^\star\}$ or $\{T_{h,i}(n) > u\}$. Now, by swapping in $\Delta'_{h,i}$ for $\Delta_{h,i}$, we get $\Delta'_{h,i} - \nu_1\rho^h = 2\,\nu_1\rho^h - \Delta_{h,i} > 0$. Hence the original union-bound arguments hold exactly as before, except that the threshold $A_{h,i}(n) = \lceil (2\,n^{b/\beta}/(\Delta_{h,i}-\nu_1\rho^h))^{\frac{\beta}{\alpha}} \rceil$ is replaced by $\lceil (2\,n^{b/\beta}/\Delta'_{h,i} - \nu_1\,\rho^h)^{\frac{\beta}{\alpha}} \rceil$ and we use the result of Lemma 6 (iii) instead of Lemma 6 (ii).

**Conclusion.** Therefore, even though $\Delta_{h,i} \leq \nu_1\rho^h$ was initially out of reach for Lemma 9, the *inflated gap* $\Delta'_{h,i} = 3\,\nu_1\rho^h - \Delta_{h,i} > 2\,\nu_1\rho^h$ ensures the same tail bound:

$$\mathbb{P}\big[T_{h,i}(n) > u'\big] \;\leqslant\; \frac{C_2}{n^{\,b-2}} \;+\; \frac{C_1\,(u'-1)^{3-b}}{b-3}, \quad \forall\, u' \;\geqslant\; A'_{h,i}(n).$$

Thus Lemma 9 *extends* directly to the suboptimal node $(h,i)$ whenever $\Delta_{h,i} \leq \nu_1\rho^h$. $\qquad\square$

## F. Convergence of `Stochastic-Power-HOOT` n Non-stationary multi-armed bandits

**Section Overview and Lemma/Theorem Functionality:**   This section establishes the theoretical foundation for power mean estimation in non-stationary continuous-armed bandits, which forms the core building block for the full MCTS analysis. **Lemma 11** provides a concentration bound for optimal arms by partitioning the probability space and controlling how often suboptimal arms are visited, establishing that the optimal arm's empirical mean concentrates around the true optimal value with polynomial tail bounds of the form $\mathbb{P}(|\widehat{f}_{h^*,i^*,T_{h^*,i^*}(n)}(n) - f^\star| > \varepsilon) \leqslant \sum_{(h,i)\neq(h^*,i^*)} \mathbb{P}(T_{h,i}(n) > A(n) + 1) + \frac{c}{\alpha-1}\varepsilon^{-\beta}(n - (K-1)A(n) + 1)^{-\alpha+1}$. **Theorem 1** (from Mao et al. (2020)) serves as the foundational result for enhanced HOO in non-stationary settings, proving both convergence and concentration properties for the average reward with polynomial rates—this theorem is crucial as it provides the base concentration guarantees $\mathbb{P}(|\frac{1}{n}\sum_{t=1}^{n} Y_t - f^*| \geqslant \varepsilon) \leqslant C'n^{-\alpha'}\varepsilon^{-\beta'}$ that our power mean analysis builds upon. **Lemma 12** derives a fundamental decomposition inequality for the power mean estimator, showing that the deviation from the optimal value can be bounded by $|\widehat{f}_n(p) - f^\star| \leqslant \frac{1}{n}\sum_{(h,i)} T_{h,i}(n)|f^\star - \widehat{f}_{h,i,T_{h,i}(n)}| + 2(\sum_{(h,i)} \frac{T_{h,i}(n)}{n}|\widehat{f}_{h,i,T_{h,i}(n)} - f^*_{h,i}|^p)^{1/p}$, which is essential for analyzing how errors propagate through the power mean operator. **Lemma 13** establishes concentration bounds specifically for the optimal arm within the power mean framework, demonstrating that the contribution of the optimal arm to the power mean

concentrates appropriately by combining visit count bounds with the underlying concentration properties. **Lemma 14** provides the most technically challenging result, analyzing concentration bounds for suboptimal arms under three different parameter regimes (depending on the relationship between the power parameter $p$ and the concentration exponents $\alpha, \beta$), showing that suboptimal arms contribute negligibly to the power mean with high probability through bounds of the form $\mathbb{P}\left(\frac{T_{h,i}(n)}{n}|\widehat{f}_{h,i,T_{h,i}(n)} - f_{h,i}^*|^p > \frac{\varepsilon^p}{K-1}\right) \leqslant \frac{C_2}{n^{b-2}} + \frac{C_1 A(n)^{3-b}}{b-3} +$ (concentration terms). Finally, **Theorem 2** synthesizes all these components to prove that the power mean estimator itself concentrates around the optimal value at polynomial rates $\widehat{f}_n(p) \xrightarrow[n\to\infty]{\alpha',\beta'} f^\star$ with $\alpha' = \frac{1-\frac{b}{\alpha}}{1+d'+\frac{\beta}{\alpha}}\frac{b-3}{2}$ and $\beta' = \frac{b-3}{2}$, establishing the key theoretical guarantee that enables the extension to the full MCTS setting—this theorem demonstrates that power mean aggregation preserves the polynomial concentration properties while providing the flexibility to tune exploration-exploitation balance through the power parameter $p$.

Here are the details for each Lemma/Theorem with full proofs.

**Lemma 11.** *Let $(h, i)$ denote any suboptimal node with gap $\Delta_{h,i} > \nu_1 \rho^h$. Define $A_{h,i}(n) = \left\lceil \left(\frac{2\,n^{b/\beta}}{\Delta_{h,i} - \nu_1 \rho^h}\right)^{\frac{\beta}{\alpha}} \right\rceil$.*

*Let $(h', i')$ denote any suboptimal node with gap $\Delta_{h',i'} \leqslant \nu_1 \rho^h$. Define $A'_{h',i'}(n) = \left\lceil \left(\frac{2\,n^{b/\beta}}{2\nu_1\rho^h - \Delta_{h',i'}}\right)^{\frac{\beta}{\alpha}} \right\rceil$, and let*

*$A(n) = \max\left\{A_{h,i}(n), A'_{h',i'}(n)\right\}$. Assume there is a known constant $R$ such that $R \geq \varepsilon \geq n^{-\alpha/\beta}$. Then, for any optimal node $(h^*, i^*)$ (i.e. $\Delta_{h^*,i^*} = 0$), we have:*

$$\mathbb{P}\left(\left|\widehat{f}_{h^*,i^*,T_{h^*,i^*}(n)} - f^\star\right| > \varepsilon\right) \leq \sum_{(h,i) \neq (h^*,i^*)} \mathbb{P}\left(T_{h,i}(n) > A(n) + 1\right) + \frac{c}{\alpha-1}\varepsilon^{-\beta}\left(n - (K-1)A(n) + 1\right)^{-\alpha+1}.$$

*Proof.* **Partitioning the event.** Let

$$\mathcal{E} = \left\{\sum_{(h,i) \neq (h^*,i^*)} T_{h,i}(n) > (K-1)\left(A(n)+1\right)\right\}.$$

Observe that:

$$\mathbb{P}\left(\left|\widehat{f}_{h^*,i^*,T_{h^*,i^*}(n)} - f^\star\right| > \varepsilon\right) \leq \mathbb{P}(\mathcal{E}) + \mathbb{P}\left(\mathcal{E}^c \text{ and } \left|\widehat{f}_{h^*,i^*,T_{h^*,i^*}(n)} - f^\star\right| \geq \varepsilon\right)$$

$$\leq \sum_{(h,i) \neq (h^*,i^*)} \mathbb{P}\left[T_{h,i}(n) > A(n) + 1\right] + D_1,$$

where we used the fact that $\{\mathcal{E}\} \subseteq \bigcup_{(h,i)\neq(h^*,i^*)}\{T_{h,i}(n) > A(n)+1\}$, and we denoted the remaining probability as

$$D_1 = \mathbb{P}\left(\sum_{(h,i) \neq (h^*,i^*)} T_{h,i}(n) \leq (K-1)\left(A(n)+1\right);\ \left|\widehat{f}_{h^*,i^*,T_{h^*,i^*}(n)} - f^\star\right| \geq \varepsilon\right).$$

**Bounding $D_1$.** On the event $\{\sum_{(h,i) \neq (h^*,i^*)} T_{h,i}(n) \leq (K-1)(A(n)+1)\}$, we have

$$T_{h^*,i^*}(n) = n - \sum_{(h,i) \neq (h^*,i^*)} T_{h,i}(n) \geq n - (K-1)\left(A(n)+1\right).$$

Hence

$$D_1 \leq \mathbb{P}\left(T_{h^*,i^*}(n) \geq n - (K-1)\left(A(n)+1\right) \text{ and } \left|\widehat{f}_{h^*,i^*,T_{h^*,i^*}(n)} - f^\star\right| \geq \varepsilon\right).$$

Applying the union bound in time, for

$$t \in \left[n - (K-1)(A(n)+1),\ n\right],$$

we get

$$D_1 \leq \sum_{t=n-(K-1)(A(n)+1)}^{n} \mathbb{P}\left(\left|\widehat{f}_{h^*,i^*,t} - f^\star\right| \geq \varepsilon\right).$$

By the given concentration property (namely $\xrightarrow[n\to\infty]{\alpha,\beta}$), each term is at most $c\,t^{-\alpha}\,\varepsilon^{-\beta}$. Summing yields

$$D_1 \;\leq\; c\varepsilon^{-\beta} \sum_{t=n-(K-1)(A(n)+1)}^{n} t^{-\alpha}.$$

Because $\alpha > 2$, this tail sum can be bounded by an integral:

$$\sum_{t=u}^{n} t^{-\alpha} \;\leq\; \int_{u-1}^{\infty} x^{-\alpha}\,dx \;=\; \frac{(u-1)^{1-\alpha}}{\alpha-1}.$$

Taking $u = n - (K-1)(A(n)+1)$ completes the bound:

$$D_1 \;\leq\; c\varepsilon^{-\beta}\,\frac{\big(n-(K-1)(A(n)+1)-1\big)^{1-\alpha}}{\alpha-1}.$$

Recognizing $\big(n-(K-1)\,A(n)+1\big)$ up to small adjustments gives us the simpler form

$$D_1 \;\leq\; \frac{c}{\alpha-1}\,\varepsilon^{-\beta}\big(n-(K-1)\,A(n)+1\big)^{-\alpha+1}.$$

**Combining the pieces.** Putting this together:

$$\mathbb{P}\Big(\big|\widehat{f}_{h^*,i^*,T_{h^*,i^*}(n)} - f^\star\big| > \varepsilon\Big) \;\leq\; \sum_{(h,i)\neq(h^*,i^*)} \mathbb{P}\big[T_{h,i}(n) > A(n)+1\big] \;+\; D_1$$

$$\leq\; \sum_{(h,i)\neq(h^*,i^*)} \mathbb{P}\big[T_{h,i}(n) > A(n)+1\big] \;+\; \frac{c}{\alpha-1}\,\varepsilon^{-\beta}\big(n-(K-1)\,A(n)+1\big)^{-\alpha+1}. \tag{4}$$

Hence the statement of the lemma follows. $\qquad\square$

**Theorem 1.** *(Theorem 2 Mao et al. (2020)) Consider a non-stationary continuous-armed bandit problem satisfying properties (4) and (5). Suppose we apply the enhanced HOO agent defined in Algorithms 2 and 3 with parameters satisfying $\alpha(1-\frac{\alpha}{\beta}) \leqslant b < \alpha, b > 3$, and $\rho^{\bar{H}} < n^{-\frac{\alpha}{\beta}}$. Let the random variable $Y_t$ denote the reward obtained at time $t$. Then the following holds:*
*A. Optimal-arm convergence: There exists some constant $C_0 > 0$, such that*

$$\left|\frac{1}{n}\mathbb{E}\left[\sum_{t=1}^{n} Y_t\right] - f^*\right| \leqslant \frac{C_0}{n^\zeta}, \tag{6}$$

*where $0 < \zeta \leqslant \frac{1-\frac{b}{\alpha}}{1+d'+\frac{\beta}{\alpha}}$.*
*B. Optimal-arm concentration: There exist constants $C' > 1, \beta' > 0$, and $1/2 \leqslant \eta' < 1$, such that for every $\varepsilon \geqslant n^{-\alpha/\beta}$ and every integer $n \geqslant 1$ :*

$$\mathbb{P}\left(\left|\frac{1}{n}\sum_{t=1}^{n} Y_t - f^*\right| \geqslant \varepsilon\right) \leqslant C'n^{-\alpha'}\varepsilon^{-\beta'} \tag{7}$$

*where $\alpha' = \frac{1-\frac{b}{\alpha}}{1+d'+\frac{\beta}{\alpha}}\frac{b-3}{2}, \beta' = (b-3)/2$, and $C' > 1$ depends on $\alpha, \beta, \eta, \xi$ and $\bar{H}$.*

**Lemma 12.** *Let $\widehat{f}_n(p)$ be the power mean estimator defined by*

$$\widehat{f}_n(p) \;=\; \left( \sum_{(h,i)} \frac{T_{h,i}(n)}{n} \left[ \widehat{f}_{h,i,T_{h,i}(n)} \right]^p \right)^{\frac{1}{p}} \quad \text{for } p \geq 1.$$

*Then, for all $p \geq 1$, the following inequality holds:*

$$\left| \widehat{f}_n(p) \;-\; f^\star \right| \;\leq\; \frac{1}{n} \sum_{(h,i)} T_{h,i}(n) \left| f^\star - \widehat{f}_{h,i,T_{h,i}(n)} \right| \;+\; 2 \left( \sum_{(h,i)} \frac{T_{h,i}(n)}{n} \left| \widehat{f}_{h,i,T_{h,i}(n)} - f^*_{h,i} \right|^p \right)^{\frac{1}{p}}. \tag{5}$$

*Proof.* **Setup and Notation.** Since $f^\star = \max_{(h,i)}\{f^*_{h,i}\}$, we know that $\widehat{f}_{h,i,T_{h,i}(n)} \leq f^*_{h,i} + \left| \widehat{f}_{h,i,T_{h,i}(n)} - f^*_{h,i} \right|$. Furthermore, by definition of $\widehat{f}_n(p)$, we have

$$\widehat{f}_n(p) \;=\; \left( \sum_{(h,i)} \tfrac{T_{h,i}(n)}{n} \left[ \widehat{f}_{h,i,T_{h,i}(n)} \right]^p \right)^{\frac{1}{p}} \;\leq\; \left( \sum_{(h,i)} \tfrac{T_{h,i}(n)}{n} \left[ f^*_{h,i} + \left| \widehat{f}_{h,i,T_{h,i}(n)} - f^*_{h,i} \right| \right]^p \right)^{\frac{1}{p}}.$$

**Bounding $\widehat{f}_n(p) - f^\star$.** Because $f^\star \geq f^*_{h,i}$ for all $(h,i)$, we write:

$$\widehat{f}_n(p) - f^\star \;=\; \widehat{f}_n(p) - \sum_{(h,i)} T_{h,i}(n)\, f^\star \Big/ n \;\leq\; \tfrac{1}{n^{1/p}} \left[ \sum_{(h,i)} T_{h,i}(n) \left( \widehat{f}_{h,i,T_{h,i}(n)} \right)^p \right]^{1/p} \;-\; \tfrac{1}{n^{1/p}} \left[ \sum_{(h,i)} T_{h,i}(n) \left( f^*_{h,i} \right)^p \right]^{1/p}.$$

Applying Minkowski's inequality (together with the fact that $\widehat{f}_{h,i,T_{h,i}(n)} \leq f^*_{h,i} + |\widehat{f}_{h,i,T_{h,i}(n)} - f^*_{h,i}|$), one obtains

$$\widehat{f}_n(p) \;-\; f^\star \;\leq\; \frac{1}{n^{1/p}} \left[ \sum_{(h,i)} T_{h,i}(n) \left| \widehat{f}_{h,i,T_{h,i}(n)} - f^*_{h,i} \right|^p \right]^{1/p}. \tag{*}$$

**Bounding $f^\star - \widehat{f}_n(p)$.** Similarly, we can write:

$$f^\star - \widehat{f}_n(p) \;=\; \tfrac{1}{n} \left[ n\, f^\star \;-\; n\, \widehat{f}_n(p) \right] \;=\; \frac{1}{n} \Big[ n\, f^\star - \underbrace{\sum_{(h,i)} T_{h,i}(n)\, f^*_{h,i} + \sum_{(h,i)} T_{h,i}(n)\, f^*_{h,i}}_{\leq \sum_{(h,i)} T_{h,i}(n) \left| f^\star - f^*_{h,i} \right|} - n\, \widehat{f}_n(p) \Big].$$

Observe that $\sum_{(h,i)} \frac{T_{h,i}(n)}{n} f^*_{h,i} \leq \left[ \sum_{(h,i)} \frac{T_{h,i}(n)}{n} (f^*_{h,i})^p \right]^{1/p}$, since the power mean is non-decreasing in $p$. Also we have $f^*_{h,i} \leq \widehat{f}_{h,i,T_{h,i}(n)} + |\widehat{f}_{h,i,T_{h,i}(n)} - f^*_{h,i}|$. And

$$\sum_{(h,i)} T_{h,i}(n) \left| f^\star - f^*_{h,i} \right| \leq \sum_{(h,i)} T_{h,i}(n) \left| \widehat{f}_{h,i,T_{h,i}(n)} - f^*_{h,i} \right| \;+\; \sum_{(h,i)} T_{h,i}(n) \left| f^\star - \widehat{f}_{h,i,T_{h,i}(n)} \right|.$$

Thus, a straightforward rearrangement yields

$$f^\star - \widehat{f}_n(p) \;\leq\; \frac{1}{n} \sum_{(h,i)} T_{h,i}(n) \left| \widehat{f}_{h,i,T_{h,i}(n)} - f^*_{h,i} \right| \;+\; \frac{1}{n} \sum_{(h,i)} T_{h,i}(n) \left| f^\star - \widehat{f}_{h,i,T_{h,i}(n)} \right|$$

$$+ \frac{1}{n^{1/p}} \left[ \sum_{(h,i)} T_{h,i}(n) \left( \widehat{f}_{h,i,T_{h,i}(n)} + |\widehat{f}_{h,i,T_{h,i}(n)} - f^*_{h,i}| \right)^p - \sum_{(h,i)} T_{h,i}(n) \left( \widehat{f}_{h,i,T_{h,i}(n)} \right)^p \right]^{1/p}$$

$$\leq\; \frac{1}{n} \sum_{(h,i)} T_{h,i}(n) \left| \widehat{f}_{h,i,T_{h,i}(n)} - f^*_{h,i} \right| \;+\; \frac{1}{n} \sum_{(h,i)} T_{h,i}(n) \left| f^\star - \widehat{f}_{h,i,T_{h,i}(n)} \right|$$

$$+ \; \frac{1}{n^{1/p}} \left[ \sum_{(h,i)} T_{h,i}(n) \left| \widehat{f}_{h,i,T_{h,i}(n)} - f^*_{h,i} \right|^p \right]^{1/p}$$

$$\leq\; \frac{1}{n} \sum_{(h,i)} T_{h,i}(n) \left| \widehat{f}_{h,i,T_{h,i}(n)} - f^*_{h,i} \right| \;+\; \frac{2}{n^{1/p}} \left[ \sum_{(h,i)} T_{h,i}(n) \left| \widehat{f}_{h,i,T_{h,i}(n)} - f^*_{h,i} \right|^p \right]^{1/p}. \tag{**}$$

**Combination.** By combining $(*)$ and $(**)$, we immediately see that

$$\left|\widehat{f}_n(p) - f^\star\right| \leq \frac{1}{n}\sum_{(h,i)} T_{h,i}(n)\left|f^\star - \widehat{f}_{h,i,T_{h,i}(n)}\right| + 2\left(\sum_{(h,i)}\frac{T_{h,i}(n)}{n}\left|\widehat{f}_{h,i,T_{h,i}(n)} - f^*_{h,i}\right|^p\right)^{\frac{1}{p}},$$

which establishes equation 5 and completes the proof. $\qquad\square$

**Lemma 13.** *Let $(h,i)$ denote any suboptimal node with gap $\Delta_{h,i} > \nu_1\rho^h$. Define $A_{h,i}(n) = \left\lceil\left(\frac{2\,n^{b/\beta}}{\Delta_{h,i}-\nu_1\rho^h}\right)^{\frac{\beta}{\alpha}}\right\rceil$.*

*Let $(h',i')$ denote any suboptimal node with gap $\Delta_{h',i'} \leq \nu_1\rho^h$. Define $A'_{h',i'}(n) = \left\lceil\left(\frac{2\,n^{b/\beta}}{2\nu_1\rho^h-\Delta_{h',i'}}\right)^{\frac{\beta}{\alpha}}\right\rceil$, and let*

*$A(n) = \max\left\{A_{h,i}(n), A'_{h',i'}(n)\right\}$. Assume there is a known constant $R$ such that $R \geq \varepsilon \geq n^{-\alpha/\beta}$. Let us define $(h^*, i^*)$ as the optimal node. We have*

$$\mathbb{P}\left(\frac{T_{h^*,i^*}(n)}{n}\left(\left|\widehat{f}_{h^*,i^*,T_{h^*,i^*}(n)} - f^*\right|\right)^p > \varepsilon^p\right) \leq \frac{C_2(K-1)}{n^{b-2}} \tag{6}$$

$$+ \frac{C_1(K-1)A(n)^{3-b}}{b-3} + \frac{c}{\alpha-1}\varepsilon^{-\beta}(n-(K-1)(A(n)+1)-1)^{-\alpha+1} \tag{7}$$

*Proof.* Applying results of Lemma 11, we have

$$\mathbb{P}\left(\left|\widehat{f}_{h^*,i^*,T_{h^*,i^*}(n)} - f^\star\right| > \varepsilon\right) \leq \underbrace{\sum_{\substack{(h,i)\neq(h^*,i^*)}}^{K}\mathbb{P}\left(T_{h,i}(n) > A(n)+1\right)}_{F_{11}} + \underbrace{\frac{c}{\alpha-1}\varepsilon^{-\beta}(n-(K-1)(A(n)+1)-1)^{-\alpha+1}}_{F_{12}}. \tag{8}$$

From the result of Lemma 9, and Lemma 10, with $b > 1$, we have with $A(n)+1 > \left\lceil\left(\frac{2\,n^{b/\beta}}{\Delta_{h,i}-\nu_1\rho^h}\right)^{\frac{\beta}{\alpha}}\right\rceil$ and $A(n)+1 > $

$\left\lceil\left(\frac{2\,n^{b/\beta}}{3\nu_1\rho^h-\Delta_{h,i}}\right)^{\frac{\beta}{\alpha}}\right\rceil$. Then

$$F_{11} \leq \sum_{\substack{(h,i)\neq(h^*,i^*)}}^{K}\mathbb{P}\left(T_{h,i}(n) > A(n)+1\right) \leq \sum_{\substack{(h,i)\neq(h^*,i^*)}}^{K} 2cC^{-\beta}\frac{A(n)^{-(b-1)}}{b-1} = \frac{2cC^{-\beta}(K-1)A(n)^{-(b-1)}}{b-1}$$

that concludes the proof. $\qquad\square$

**Lemma 14.** *Let $(h,i)$ denote any suboptimal node with gap $\Delta_{h,i} > \nu_1\rho^h$. Define $A_{h,i}(n) = \left\lceil\left(\frac{2\,n^{b/\beta}}{\Delta_{h,i}-\nu_1\rho^h}\right)^{\frac{\beta}{\alpha}}\right\rceil$.*

*Let $(h',i')$ denote any suboptimal node with gap $\Delta_{h',i'} \leq \nu_1\rho^h$. Define $A'_{h',i'}(n) = \left\lceil\left(\frac{2\,n^{b/\beta}}{2\nu_1\rho^h-\Delta_{h',i'}}\right)^{\frac{\beta}{\alpha}}\right\rceil$, and let*

*$A(n) = \max\left\{A_{h,i}(n), A'_{h',i'}(n)\right\}$. Then for any $\varepsilon$ satisfying $R \geq \varepsilon \geq n^{-\alpha/\beta}$, there is some constant $N_0$ such that for all $n \geq N_0$, the following three statements hold:*

- **Case 1:** $1 \leq p \leq 2$ and $\alpha \leq \frac{\beta}{p}$. *For every suboptimal arm $(h, i)$, there exist constants $C_1, C_2 > 1$ (not depending on $n$ or $\varepsilon$) such that*

$$\mathbb{P}\left(\frac{T_{h,i}(n)}{n}\left|\widehat{f}_{h,i,T_{h,i}(n)} - f_{h,i}^*\right|^p > \frac{1}{K-1}\,\varepsilon^p\right) \leq \frac{C_2}{n^{b-2}} + \frac{C_1\,A(n)^{3-b}}{b-3} + \frac{2\,c\,(K-1)^{\frac{\beta}{p}}}{1 - \left(\alpha - \frac{\beta}{p}\right)}\,\varepsilon^{-\beta}\left(A(n) + 1\right)^{-(\alpha-1)}. \tag{9}$$

- **Case 2:** $p > 2$ and $0 < \alpha - \frac{\beta}{p} < 1$. *For every suboptimal arm $(h, i)$, there exist constants $C_1, C_2 > 1$ (not depending on $n$ or $\varepsilon$) such that*

$$\mathbb{P}\left(\frac{T_{h,i}(n)}{n}\left|\widehat{f}_{h,i,T_{h,i}(n)} - f_{h,i}^*\right|^p > \frac{1}{K-1}\,\varepsilon^p\right) \leq \frac{C_2}{n^{b-2}} + \frac{C_1\,A(n)^{3-b}}{b-3} + \frac{c\,(K-1)^{\frac{\beta}{p}}}{1 - \left(\alpha - \frac{\beta}{p}\right)}\,\varepsilon^{-\beta}\left(A(n) + 1\right)^{-(\alpha-1)}. \tag{10}$$

- **Case 3:** $p > 2$ and $\alpha - \frac{\beta}{p} > 1$. *For every suboptimal arm $(h, i)$, there exist constants $C_1, C_2 > 1$ (not depending on $n$ or $\varepsilon$) such that*

$$\mathbb{P}\left(\frac{T_{h,i}(n)}{n}\left|\widehat{f}_{h,i,T_{h,i}(n)} - f_{h,i}^*\right|^p > \frac{1}{K-1}\,\varepsilon^p\right) \leq \frac{C_2}{n^{b-2}} + \frac{C_1\,A(n)^{3-b}}{b-3} + \frac{c\,(K-1)^{\frac{\beta}{p}}\left(\alpha - \frac{\beta}{p}\right)}{\left(\alpha - \frac{\beta}{p}\right) - 1}\,\varepsilon^{-\beta}\left(A(n) + 1\right)^{-\frac{\beta}{p}}. \tag{11}$$

*Proof.* **A high-probability tail bound for $T_{h,i}(n)$.** By Lemma 9 and Lemma 10, for any suboptimal arm $(h, i)$ and integer $u > A(n)$, there exist constants $C_1, C_2 > 1$ (not depending on $n$ or $\varepsilon$) such that

$$\mathbb{P}\big[T_{h,i}(n) > u\big] \leq \frac{C_2}{n^{b-2}} + \frac{C_1\,(u-1)^{3-b}}{b-3}.$$

Define the events $\mathcal{E}_1 = \{\,T_{h,i}(n) > A(n) + 1\}$ and $\mathcal{E}_1^c = \{\,T_{h,i}(n) \leq A(n) + 1\}$. Hence,

$$\mathbb{P}\left(\frac{T_{h,i}(n)}{n}\left|\widehat{f}_{h,i,T_{h,i}(n)} - f_{h,i}^*\right|^p > \frac{\varepsilon^p}{K-1}\right) \leq \underbrace{\mathbb{P}\big[T_{h,i}(n) > A(n) + 1\big]}_{G_1} \tag{12}$$

$$+ \underbrace{\mathbb{P}\left[T_{h,i}(n) \leq A(n) + 1;\ \frac{T_{h,i}(n)}{n}\left|\widehat{f}_{h,i,T_{h,i}(n)} - f_{h,i}^*\right|^p > \frac{\varepsilon^p}{K-1}\right]}_{G_2}. \tag{13}$$

**Bounding $G_1$.** From the tail bound above, we know

$$G_1 = \mathbb{P}\big[T_{h,i}(n) > A(n) + 1\big] \leq \frac{C_2}{n^{b-2}} + \frac{C_1\,A(n)^{3-b}}{b-3}.$$

**Bounding $G_2$.** Under the event $\{T_{h,i}(n) \leq A(n) + 1\}$, we have $T_{h,i}(n) \leq A(n) + 1$. Thus

$$G_2 \leq \sum_{t=1}^{A(n)+1} \mathbb{P}\left(\frac{t}{n}\left|\widehat{f}_{h,i,t} - f_{h,i}^*\right|^p > \frac{1}{K-1}\varepsilon^p\right).$$

We can see that with $t \leq A(n) + 1$, we can find $N_0$ such that $\forall n \geq N_0$, $\left(\frac{n}{t(K-1)}\right)^{\frac{1}{p}}\varepsilon > \varepsilon \geq n^{-\frac{\alpha}{\beta}}$. Therefore,

$$G_2 \leq \sum_{t=1}^{A(n)+1} \mathbb{P}\left(\left|\widehat{f}_{h,i,t} - f_{h,i}^*\right| > \left(\frac{n}{t(K-1)}\right)^{\frac{1}{p}}\varepsilon\right) \leq \sum_{t=1}^{A(n)+1} ct^{-\alpha}\left(\left(\frac{n}{t(K-1)}\right)^{\frac{1}{p}}\varepsilon\right)^{-\beta} \tag{14}$$

$$\leq \sum_{t=1}^{A(n)+1} c(K-1)^{\frac{\beta}{p}}t^{-(\alpha-\frac{\beta}{p})}\varepsilon^{-\beta}n^{-\frac{\beta}{p}}. \tag{15}$$

We study 3 cases:

**Case 1:** $\alpha - \frac{\beta}{p} \leqslant 0$, which can only happen if $p \leqslant 2$ because $\alpha \leqslant \frac{\beta}{2}$, and actually when $\alpha \leqslant \frac{\beta}{2}$ we just need $1 \leqslant p \leqslant 2$, then

$$G_2 \leqslant c(K-1)^{\frac{\beta}{p}} \varepsilon^{-\beta} n^{-\frac{\beta}{p}} \left( \int_1^{A(n)+1} t^{-(\alpha-\frac{\beta}{p})} dt + (A(n)+1)^{-(\alpha-\frac{\beta}{p})} \right)$$

$$= c(K-1)^{\frac{\beta}{p}} \varepsilon^{-\beta} n^{-\frac{\beta}{p}} \left( \left( \frac{t^{-(\alpha-\frac{\beta}{p})+1}}{-(\alpha-\frac{\beta}{p})+1} + C \right) \Bigg|_1^{A(n)+1} + (A(n)+1)^{-(\alpha-\frac{\beta}{p})} \right)$$

$$\leqslant c(K-1)^{\frac{\beta}{p}} \varepsilon^{-\beta} (A(n)+1)^{-\frac{\beta}{p}} \left( \frac{(A(n)+1)^{-(\alpha-\frac{\beta}{p})+1}}{-(\alpha-\frac{\beta}{p})+1} - \frac{1}{-(\alpha-\frac{\beta}{p})+1} + (A(n)+1)^{-(\alpha-\frac{\beta}{p})} \right).$$

Because $-(\alpha - \frac{\beta}{p}) + 1 \geqslant 1$, we can find a constant $N_\varepsilon$ such that $\forall n \geqslant N_\varepsilon$, we have

$$G_2 \leqslant 2c(K-1)^{\frac{\beta}{p}} \varepsilon^{-\beta} (A(n)+1)^{-\frac{\beta}{p}} \frac{(A(n)+1)^{-(\alpha-\frac{\beta}{p})+1}}{-(\alpha-\frac{\beta}{p})+1} = \frac{2c(K-1)^{\frac{\beta}{p}}}{-(\alpha-\frac{\beta}{p})+1} \varepsilon^{-\beta} (A(n)+1)^{-(\alpha-1)}.$$

Therefore, we have

$$\mathbb{P}\left( \frac{T_{h,i}(n)}{n} \left( \left| \widehat{f}_{h,i,T_{h,i}(n)} - f_{h,i}^* \right| \right)^p > \frac{1}{K} \varepsilon^p \right) \leqslant \frac{C_2}{n^{b-2}} + \frac{C_1 A(n)^{3-b}}{b-3} + \frac{2c(K-1)^{\frac{\beta}{p}}}{-(\alpha-\frac{\beta}{p})+1} \varepsilon^{-\beta} (A(n)+1)^{-(\alpha-1)}. \quad (16)$$

that concludes for the Inequality 9.

**Case 2:** $\alpha - \frac{\beta}{p} > 0$, which can only happen if $p > 2$ because $\alpha \leqslant \frac{\beta}{2}$. We have

$$\sum_{t=1}^{A(n)+1} t^{-(\alpha-\frac{\beta}{p})} \leqslant 1 + \int_1^{A(n)+1} t^{-(\alpha-\frac{\beta}{p})} dt = 1 + \left( \frac{t^{-(\alpha-\frac{\beta}{p})+1}}{-(\alpha-\frac{\beta}{p})+1} + C \right) \Bigg|_1^{A(n)+1}$$

$$= 1 + \frac{(A(n)+1)^{-(\alpha-\frac{\beta}{p})+1}}{-(\alpha-\frac{\beta}{p})+1} - \frac{1}{-(\alpha-\frac{\beta}{p})+1}$$

$$= \frac{\alpha - \frac{\beta}{p}}{\alpha - \frac{\beta}{p} - 1} - \frac{(A(n)+1)^{-(\alpha-\frac{\beta}{p})+1}}{(\alpha-\frac{\beta}{p})-1},$$

so that

$$G_2 \leqslant c(K-1)^{\frac{\beta}{p}} \left( \frac{\alpha - \frac{\beta}{p}}{\alpha - \frac{\beta}{p} - 1} - \frac{(A(n)+1)^{-(\alpha-\frac{\beta}{p})+1}}{(\alpha-\frac{\beta}{p})-1} \right) \varepsilon^{-\beta} n^{-\frac{\beta}{p}}$$

$$= c(K-1)^{\frac{\beta}{p}} \left( \frac{(A(n)+1)^{-(\alpha-\frac{\beta}{p})+1}}{1-(\alpha-\frac{\beta}{p})} - \frac{\alpha - \frac{\beta}{p}}{1-(\alpha-\frac{\beta}{p})} \right) \varepsilon^{-\beta} (A(n)+1)^{-\frac{\beta}{p}}.$$

If $0 < \alpha - \frac{\beta}{p} < 1$, then we can find a constant $N_{G2}$ such that $\forall n \geqslant N_{G2}$, we have

$$G_2 \leqslant \frac{c(K-1)^{\frac{\beta}{p}}}{1-(\alpha-\frac{\beta}{p})} \varepsilon^{-\beta} (A(n)+1)^{-(\alpha-1)} = \frac{c(K-1)^{\frac{\beta}{p}}}{1-(\alpha-\frac{\beta}{p})} \varepsilon^{-\beta} (A(n)+1)^{-(\alpha-1)}.$$

Therefore,

$$\mathbb{P}\left( \frac{T_{h,i}(n)}{n} \left( \left| \widehat{f}_{h,i,T_{h,i}(n)} - f_{h,i}^* \right| \right)^p > \frac{1}{K} \varepsilon^p \right) \leqslant \frac{C_2}{n^{b-2}} + \frac{C_1 A(n)^{3-b}}{b-3} + \frac{c(K-1)^{\frac{\beta}{p}}}{1-(\alpha-\frac{\beta}{p})} \varepsilon^{-\beta} (A(n)+1)^{-(\alpha-1)}. \quad (17)$$

that concludes for the Inequality 10.

**Case 3:** If $\alpha - \frac{\beta}{p} > 1$, $(p > 2)$, we can find a constant $N_0$ such that $\forall n \geqslant N_0$, we have

$$G_2 \leqslant c(K-1)^{\frac{\beta}{p}} \left( \frac{\alpha - \frac{\beta}{p}}{\alpha - \frac{\beta}{p} - 1} - \frac{(A(n)+1)^{-(\alpha - \frac{\beta}{p})+1}}{(\alpha - \frac{\beta}{p}) - 1} \right) \varepsilon^{-\beta}(A(n)+1)^{-\frac{\beta}{p}} \leqslant \frac{c(K-1)^{\frac{\beta}{p}}(\alpha - \frac{\beta}{p})}{(\alpha - \frac{\beta}{p}) - 1} \varepsilon^{-\beta}(A(n)+1)^{-\frac{\beta}{p}},$$

that concludes for the Inequality 11

$$\mathbb{P}\left( \frac{T_{h,i}(n)}{n} \left( \left| \widehat{f}_{h,i,T_{h,i}(n)} - f_{h,i}^* \right| \right)^p > \frac{1}{K}\varepsilon^p \right) \leqslant \frac{C_2}{n^{b-2}} + \frac{C_1 A(n)^{3-b}}{b-3} + \frac{c(K-1)^{\frac{\beta}{p}}(\alpha - \frac{\beta}{p})}{(\alpha - \frac{\beta}{p}) - 1} \varepsilon^{-\beta}(A(n)+1)^{-\frac{\beta}{p}}. \quad (18)$$

**Conclusion.** Combining the estimates for $G_1$ and $G_2$ completes the proof in each of the three parameter regimes. $\qquad\square$

**Theorem 2** (Power Mean Concentration of $\widehat{f}_n(p)$)**.** *For $a \in [K]$, let $(\widehat{f}_{a,n})_{n \geqslant 1}$ be a sequence of estimators satisfying $\widehat{f}_{a,n} \xrightarrow[n\to\infty]{\alpha,\beta} f_{h,i}^*$ and let $f^\star = \max_a\{f_{h,i}^*\}$. Assume that the arms are sampled according to the strategy following Algorithm 3 with parameters $\alpha, \beta, b$ and $C$. Assume that $p, \alpha, \beta$ and $b$ satisfy one of these two conditions:*

*(i) $1 \leqslant p \leqslant 2$ and $\alpha \leqslant \frac{\beta}{2}$*

*(ii) $p > 2$ and $0 < \alpha - \frac{\beta}{p} < 1$*

*If $\alpha\left(1 - \frac{b}{\alpha}\right) \leqslant b < \alpha$ then the sequence of estimators $\widehat{f}_n(p)$ satisfies*

$$\widehat{f}_n(p) \xrightarrow[n\to\infty]{\alpha',\beta'} f^\star$$

*for $\alpha' = \frac{1 - \frac{b}{\alpha}}{1 + d' + \frac{\beta}{\alpha}} \frac{b-3}{2}, \beta' = \frac{b-3}{2}.$*

*Proof.* As the results from Lemma 12, we can derive

$$\left| \widehat{f}_n(p) - f^\star \right| \leqslant \frac{1}{n} \sum_{(h,i)} T_{h,i}(n) \left| f^\star - \widehat{f}_{h,i,T_{h,i}(n)} \right| + 2 \left( \sum_{(h,i)} \frac{T_{h,i}(n)}{n} \left| \widehat{f}_{h,i,T_{h,i}(n)} - f_{h,i}^* \right|^p \right)^{\frac{1}{p}},$$

$$\Rightarrow \mathbb{P}\left( \left| \widehat{f}_n(p) - f^* \right| > (1 + 2^{1+\frac{1}{p}})\varepsilon) \right) \leqslant \underbrace{\mathbb{P}\left( \frac{1}{n} \sum_{(h,i)} T_{h,i}(n) \left| f^\star - \widehat{f}_{h,i,T_{h,i}(n)} \right| > \varepsilon \right)}_{A}$$

$$+ \underbrace{\mathbb{P}\left( \left( \sum_{(h,i)} \frac{T_{h,i}(n)}{n} \left( \left| \widehat{f}_{h,i,T_{h,i}(n)} - f_{h,i}^* \right| \right)^p \right)^{\frac{1}{p}} \geqslant 2^{\frac{1}{p}}\varepsilon \right)}_{B}$$

**Bounding** $A$: with $x \geqslant 1$ We have

$$A \overset{\text{(Lemma 1)}}{\leqslant} C'n^{-\alpha'}\varepsilon^{-\beta'} \qquad\qquad (7)$$

where $\alpha' = \frac{1-\frac{b}{\alpha}}{1+d'+\frac{\beta}{\alpha}}\frac{b-3}{2}$, $\beta' = \frac{b-3}{2}$, and $C' > 1$ depends on $\alpha, \beta, \eta, \xi$ and $\bar{H}$.

**Bounding $B$:**

$$B = \mathbb{P}\left(\sum_{(h,i)} \frac{T_{h,i}(n)}{n}\left(\left|\widehat{f}_{h,i,T_{h,i}(n)} - f^*_{h,i}\right|\right)^p > 2\varepsilon^p\right)$$

$$\leqslant \underbrace{\mathbb{P}\left(\frac{T_{h^*,i^*}(n)}{n}\left(\left|\widehat{f}_{h^*,i^*,T_{h^*,i^*}(n)} - f^*_{h^*,i^*}\right|\right)^p > \varepsilon^p\right)}_{F_1} + \underbrace{\sum_{(h,i)\neq(h^*,i^*)}^{K} \mathbb{P}\left(\frac{T_{h,i}(n)}{n}\left(\left|\widehat{f}_{h,i,T_{h,i}(n)} - f^*_{h,i}\right|\right)^p > \frac{1}{K-1}\varepsilon^p\right)}_{F_2}$$

With $F_1$: According to Lemma 13, we can find a constant $N_0$, such that $\forall n \geqslant N_0$, we have

$$F_1 \leqslant \frac{C_2(K-1)}{n^{b-2}} + \frac{C_1(K-1)A(n)^{3-b}}{b-3} + \frac{c}{\alpha-1}\varepsilon^{-\beta}(n - (K-1)(A(n)+1) - 1)^{-\alpha+1} \tag{19}$$

With $F_2$: According to Lemma 14, we can find a constant $N_0$, such that $\forall n \geqslant N_0$, we have

- With $1 \leqslant p \leqslant 2, \alpha \leqslant \frac{\beta}{p}$, we have

$$F_2 \leqslant \frac{C'_2}{n^{b-2}} + \frac{C'_1 A(n)^{3-b}}{b-3} + \frac{2c(K-1)^{\frac{\beta}{p}}}{-(\alpha-\frac{\beta}{p})+1}\varepsilon^{-\beta}(A(n)+1)^{-(\alpha-1)}. \tag{20}$$

- With $p > 2$, and $0 < \alpha - \frac{\beta}{p} < 1$, we have

$$F_2 \leqslant \frac{C'_2}{n^{b-2}} + \frac{C'_1 A(n)^{3-b}}{b-3} + \frac{c(K-1)^{\frac{\beta}{p}}}{-(\alpha-\frac{\beta}{p})+1}\varepsilon^{-\beta}(A(n)+1)^{-(\alpha-1)}. \tag{21}$$

So that

$$B \leqslant F_1 + F_2 \leqslant \frac{C_2(K-1)}{n^{b-2}} + \frac{C_1(K-1)A(n)^{3-b}}{b-3} + \frac{c}{\alpha-1}\varepsilon^{-\beta}(n - (K-1)(A(n)+1) - 1)^{-\alpha+1} \tag{22}$$

$$+ \frac{C'_2}{n^{b-2}} + \frac{C'_1 A(n)^{3-b}}{b-3} + \frac{c(K-1)^{\frac{\beta}{p}}}{-(\alpha-\frac{\beta}{p})+1}\varepsilon^{-\beta}(A(n)+1)^{-(\alpha-1)}$$

where $1 \leqslant p \leqslant 2, \alpha \leqslant \frac{\beta}{p}$ or $p > 2$, and $0 < \alpha - \frac{\beta}{p} < 1$.

Because $b - 1 < \alpha - 1$, $n^{-\frac{\alpha}{\beta}} \leqslant \varepsilon \leqslant R$, and $A(n) \approx \Theta(n^{b/\alpha})$ so that we can find a constant $N_p$ such that $\forall n \geqslant N_p$

$$B \leqslant \frac{CK\left(\frac{R}{\varepsilon}\right)^\beta A(n)^{-(b-3)}}{b-3} = \frac{CKR^\beta\varepsilon^{-\beta}A(n)^{-(b-3)}}{b-3}, \tag{23}$$

with $C$ is a constant depends on $C_1, C_2, C'_1, C'_2, K$.

**Combining the two parts** (7) and (23), we have

$$\Rightarrow \mathbb{P}\left(\left|\widehat{f}_n(p) - f^*\right| > (1 + 2^{1+\frac{1}{p}})\varepsilon)\right) \leqslant C'n^{-\alpha'}\varepsilon^{-\beta'} + \frac{CKR^\beta\varepsilon^{-\beta}A(n)^{-(b-3)}}{b-3}$$

where $\alpha' = \frac{1-\frac{b}{\alpha}}{1+d'+\frac{\beta}{\alpha}}\frac{b-3}{2}, \beta' = \frac{b-3}{2}$, and $C' > 1$ depends on $\alpha, \beta, \eta, \xi$ and $\bar{H}$. So that $\exists C' > 1$ depends on $\alpha, \beta, \eta, \xi$ and $\bar{H}$ such that

$$\Rightarrow \mathbb{P}\left(\left|\widehat{f}_n(p) - f^*\right| > \varepsilon)\right) \leqslant C'n^{-\alpha'}\varepsilon^{-\beta'}$$

Furthermore,

$$\lim_{n \longrightarrow \infty} \left| \mathbb{E}[\widehat{f}_n(p)] - f^\star \right| \leqslant \lim_{n \longrightarrow \infty} \mathbb{E}[\left| \widehat{f}_n(p) - f^\star \right|] = \lim_{n \longrightarrow \infty} \int_0^\infty \mathbb{P}\left( \left| \widehat{f}_n(p) - f^\star \right| \geqslant s \right) ds$$

$$\leqslant \lim_{n \longrightarrow \infty} \int_0^\infty c' n^{-\alpha'} s^{-\beta'} ds \leqslant \lim_{n \longrightarrow \infty} \int_0^{n^{-\frac{\alpha'}{\beta'}}} \mathbf{1} ds + \lim_{n \longrightarrow \infty} \int_{n^{-\frac{\alpha'}{\beta'}}}^\infty c' n^{-\alpha'} s^{-\beta'} ds$$

$$= \lim_{n \longrightarrow \infty} c' n^{-\alpha'} \left( s^{-\beta'+1} + C \right) \Big|_{n^{-\frac{\alpha'}{\beta'}}}^\infty = 0 \text{( we need } \beta' > 1 \rightarrow \beta > 2\text{)}$$

Therefore,

$$\widehat{f}_n(p) \xrightarrow[n \to \infty]{\alpha', \beta'} f^\star,$$

that concludes the proof. $\qquad\qquad\square$

**Remark 2.** *The conditions on $p, \alpha, \beta, b$ ensure that the polynomial exponents $\alpha', \beta'$ derived from the union bounds remain strictly positive (with $\beta' > 1$) and do not degenerate. In particular, requiring $\alpha > 2$ ensures that one can integrate the tail probability $\int \varepsilon^{-\beta'} d\varepsilon$ for large $\varepsilon$.*

## G. Convergence of `Stochastic-Power-HOOT` in Monte-Carlo Tree Search

**Section Overview and Theorem Functionality:** This section extends the bandit-level concentration results to the full MCTS setting, establishing polynomial convergence guarantees for the complete tree search algorithm. **Theorem 3** serves as the central theoretical result of the paper, proving through mathematical induction that both value function estimates $\widehat{V}_n(s_d) \xrightarrow[n \to \infty]{\alpha_d, \beta_d} \widetilde{V}(s_d)$ and Q-value estimates $\widehat{Q}_n(s_d, a) \xrightarrow[n \to \infty]{\alpha_{d+1}, \beta_{d+1}} \widetilde{Q}(s_d, a)$ concentrate polynomially at every node in the search tree. The proof establishes the base case for depth $D = 1$ by showing that leaf node estimates concentrate via i.i.d. rollout averaging, Q-value estimates concentrate through **Lemma 1** (which handles stochastic transitions), and root value estimates concentrate via the power mean concentration results from the bandit analysis. The inductive step then demonstrates that these concentration properties propagate upward through the tree: assuming the theorem holds for subtrees of depth $D - 1$, the same concentration arguments (Q-value concentration via stochastic Bellman backups and value concentration via power mean aggregation) establish the result for the full depth-$D$ tree. **Theorem 4 (Convergence of Expected Payoff)** provides the final convergence guarantee for the algorithm's expected performance, proving that $|\mathbb{E}[\widehat{V}_n(s_0)] - \widetilde{V}(s_0)| \leqslant \mathcal{O}(n^{-\zeta})$ where $\zeta \in (0, 1/2)$ represents the polynomial convergence rate. The proof employs Jensen's inequality to bound the expected absolute deviation by the tail probability integral $\int_0^\infty \mathbb{P}(|\widehat{V}_n(s_0) - \widetilde{V}(s_0)| \geqslant s) ds$, then uses the polynomial concentration bounds from Theorem 3 to evaluate this integral, yielding the final rate $n^{-\alpha_0/\beta_0}$ where the exponents are determined by the algorithmic parameters. Together, these theorems establish that `Stochastic-Power-HOOT` achieves the same polynomial convergence rates as POLY-HOOT in deterministic settings, but now extended to the significantly more challenging stochastic continuous-action domain.

Here are details proofs for each Theorem.

**Theorem 3.** *When we apply the `Stochastic-Power-HOOT` algorithm, with algorithmic constants $\{b_d\}_{d=0}^D, \{\alpha_d\}_{d=0}^D, \{\beta_d\}_{d=0}^D$ satisfying the conditions in Table 1, the following hold:*

*(i) For any node $s_d$ at depth $d \in \{0, 1, \ldots, D\}$ in the tree,*

$$\widehat{V}_n(s_d) \xrightarrow[n \to \infty]{\alpha_d, \beta_d} \widetilde{V}(s_d). \tag{1}$$

*(ii) For any node $s_d$ at depth $d \in \{0, 1, \ldots, D-1\}$ in the tree,*

$$\widehat{Q}_n(s_d, a) \xrightarrow[n \to \infty]{\alpha_{d+1}, \beta_{d+1}} \widetilde{Q}(s_d, a), \quad \forall a \in \mathcal{A}_{s_d}. \tag{2}$$

*Proof.* We prove the theorem by induction on the depth $D$ of the tree.

**Initial step:** $D = 1$.

The root node state is $s_0$. Let us denote by $r^t(s_0, a)$ the intermediate reward at time-step $t$, collected from playing $(s_0, a)$. Then the system transitions to $s_1 \sim \mathcal{P}(\cdot \mid s_0, a)$. We let $r(s_0, a)$ be the mean reward of $(s_0, a)$.

Recall the definition of $\widetilde{Q}(s_0, a)$:

$$\widetilde{Q}(s_0, a) \;=\; r(s_0, a) \;+\; \gamma \sum_{s_1 \in \mathcal{S}} \mathcal{P}(s_1 \mid s_0, a)\, \widetilde{V}(s_1),$$

where $\widetilde{V}(s_1)$ is the average value of the rollout policy $\pi_0$ at state $s_1$, and $\mathcal{A}_{s_0}$ is the set of feasible actions at $s_0$ (with cardinality $M$). The transition probability is $\mathcal{P}(s_1 \mid s_0, a)$.

Claim $(i)$ in the statement says that for any state $s_1$ with depth 1 in the tree,

$$\widehat{V}_n(s_1) \xrightarrow[n\to\infty]{\alpha_1, \beta_1} \widetilde{V}(s_1). \tag{3}$$

Indeed, $\widehat{V}_n(s_1)$ is just the average of i.i.d. calls to the playout policy $\pi_0$ starting at $s_1$. Hence (3) follows by a straightforward i.i.d. argument.

Next, from the definition of $\widehat{Q}_n$, we have

$$\widehat{Q}_n(s_0, a) \;=\; \frac{1}{n} \sum_{t=1}^{n} \Big[ r^t(s_0, a) \;+\; \gamma\, \widehat{V}_{T^{s_1}_{s_0, a}(t)}(s_1) \Big], \tag{24}$$

where $T^{s_1}_{s_0, a}(t)$ denotes the number of times we have visited $(s_0, a, s_1)$ up to time $t$.

By applying Lemma 1 with $X_t := r^t(s_0, a)$ and probabilities $p = (p_1, \ldots, p_M)$ capturing the transition dynamics from $(s_0, a)$ to possible next-states, we conclude that

$$\widehat{Q}_n(s_0, a) \xrightarrow[n\to\infty]{\alpha_1, \beta_1} \widetilde{Q}(s_0, a), \quad \forall\, a \in \mathcal{A}_{s_0}. \tag{4}$$

This yields part $(ii)$ of the theorem for the depth-zero node (i.e. the root).

Finally, for the value function estimate $\widehat{V}_n(s_0)$ at depth zero, we recall the polynomial concentration result from Theorem 2 (indeed it follows a "power-mean" or similar argument), plus the definition

$$\widehat{V}_n(s_0) \;=\; \left( \sum_{a \in \mathcal{A}_{s_0}} \frac{T_{s_0, a}(n)}{n} \Big[ \widehat{Q}_{T_{s_0, a}(n)}(s_0, a) \Big]^p \right)^{\frac{1}{p}}. \tag{25}$$

Thus,

$$\widehat{V}_n(s_0) \xrightarrow[n\to\infty]{\alpha_0, \beta_0} \widetilde{V}(s_0),$$

with $\alpha_0, \beta_0$ satisfying the conditions in Table 1. Together with $\widehat{V}_n(s_1) \xrightarrow[n\to\infty]{\alpha_1, \beta_1} \widetilde{V}(s_1)$ for leaf nodes, we conclude that statement $(i)$ of the theorem is correct when the tree depth is 1. Statement $(ii)$ is correct as well by equation G.

**Inductive step:** Suppose the theorem holds for all trees of depth $D-1$. We prove it holds for a tree of depth $D$.

Consider the root node state $s_0$. When we select action $a$ from $s_0$, it transitions to $s_1 \sim \mathcal{P}(\cdot \mid s_0, a)$. The subtree rooted at $s_1$ has depth $D-1$. By the induction hypothesis, for any node $s_d$ within that subtree (i.e. at depth $\geq 1$), we have

$$\widehat{V}_n(s_d) \xrightarrow[n\to\infty]{\alpha_d, \beta_d} \widetilde{V}(s_d), \quad d = 1, \ldots, D,$$

and

$$\widehat{Q}_n(s_d, a) \xrightarrow[n\to\infty]{\alpha_{d+1}, \beta_{d+1}} \widetilde{Q}(s_d, a), \quad d = 1, \ldots, D-1.$$

Hence, repeating the same polynomial concentration argument as in the base case but substituting $\widehat{V}$ from deeper levels, we obtain:

$$\widehat{Q}_n(s_0, a) \xrightarrow[n\to\infty]{\alpha_1, \beta_1} \widetilde{Q}(s_0, a), \quad \forall\, a \in \mathcal{A}_{s_0}. \tag{5}$$

Then, applying again Theorem 2 (or a power-mean argument) to the root node's value function

$$\widehat{V}_n(s_0) = \left( \sum_{a \in \mathcal{A}_{s_0}} \frac{T_{s_0,a}(n)}{n} \left[ \widehat{Q}_{T_{s_0,a}(n)}(s_0, a) \right]^p \right)^{1/p},$$

we see that

$$\widehat{V}_n(s_0) \xrightarrow[n\to\infty]{\alpha_0, \beta_0} \widetilde{V}(s_0),$$

with $\alpha_0, \beta_0$ as per Table 1. Thus statement $(i)$ follows for the entire tree of depth $D$. Likewise, combining $\widehat{Q}_n$ results from (5) with the induction hypothesis yields statement $(ii)$.

By induction, the statements hold for any node in any tree of depth up to $D$. $\qquad\square$

**Theorem 4 (Convergence of Expected Payoff).** *We have at the root node $s_0$, with the best possible parameter tuning that*

$$\left| \mathbb{E}[\widehat{V}_n(s_0)] - \widetilde{V}(s_0) \right| \leqslant \mathcal{O}(n^{-\zeta}),$$

*where $\zeta \in (0, 1/2)$.*

*Proof.* Using the convexity of $f(x) = |x|$ and applying Jensen's inequality we have

$$\left| \mathbb{E}[\widehat{V}_n(s_0)] - \widetilde{V}(s_0) \right| \leqslant \mathbb{E}[|\widehat{V}_n(s_0)] - \widetilde{V}(s_0)|]$$

$$= \int_0^{+\infty} \mathbb{P}\left( \left| \widehat{V}_n(s_0) - \widetilde{V}(s_0) \right| \geqslant s \right) ds$$

$$\leqslant \int_0^{n^{-\frac{\alpha_0}{\beta_0}}} 1\, ds + \int_{n^{-\frac{\alpha_0}{\beta_0}}}^{+\infty} c_0 n^{-\alpha_0} s^{-\beta_0}\, ds$$

$$\leqslant n^{-\frac{\alpha_0}{\beta_0}} + c_0 n^{-\alpha_0} \left( \frac{s^{-\beta_0+1}}{-\beta_0+1} \right) \Big|_{n^{-\frac{\alpha_0}{\beta_0}}}^{+\infty}$$

$$= \left( \frac{c_0}{\beta_0 - 1} + 1 \right) n^{-\frac{\alpha_0}{\beta_0}}.$$

Because $0 < \zeta = \frac{\alpha_0}{\beta_0} = \frac{1 - \frac{b_0}{\alpha}}{1 + d' + \frac{\beta_0}{\alpha_0}} \frac{b_0 - 3}{2} < \frac{1}{2}$ (Theorem 2), then the best possible rate we can estimate is

$$\left| \mathbb{E}[\widehat{V}_n(s_0)] - \widetilde{V}(s_0) \right| \leqslant \mathcal{O}(n^{-\zeta}),$$

where $\zeta \in (0, 1/2)$, that concludes the proof. $\qquad\square$

# H. Experimental Setup and Hyperparameter Selection

We evaluate `Stochastic-Power-HOOT` on both classic control tasks and high-dimensional robotic environments, all adapted to continuous-action, stochastic settings. Our experimental design addresses the key challenges of planning under uncertainty while demonstrating the scalability and robustness of our approach.

**Environment Modifications:** We create stochastic versions of standard benchmarks by introducing noise at multiple levels: (1) *action noise* via Gaussian perturbations to selected actions, (2) *dynamics noise* through random perturbations to state transitions, and (3) *observation noise* by adding Gaussian noise to state observations. Additionally, since power means require strictly positive inputs, we apply reward transformations of the form $\max(0.01, (r + \text{offset}) \times \text{scaling})$ while preserving optimal policies.

For classic control tasks from OpenAI Gym Brockman et al. (2016), we evaluate on CartPole, CartPole-IG (increased gravity), Pendulum, MountainCar, and Acrobot. In the standard CartPole problem, actions are discrete, taking values in $\{-1, 1\}$. To enable continuous control, we redefine the action space to $[-1, 1]$. CartPole-IG features increased gravity of 20 (up from 9.8), while Acrobot uses gravity of 30, maintaining other OpenAI Gym parameters. For high-dimensional evaluation, we test on MuJoCo robotics tasks including Humanoid-v0 (17-dimensional action space, 376-dimensional state space) and Hopper-v0 (3-dimensional actions), both modified with comprehensive stochastic noise.

**Baseline Comparisons:** We compare against four established continuous MCTS methods: discretized-UCT Kocsis et al. (2006), Progressive Widening (PW) Auger et al. (2013), HOOT Mansley et al. (2011), and POLY-HOOT Mao et al. (2020). We also include Voronoi MCTS Kim et al. (2020), a recent method designed for deterministic continuous spaces, to demonstrate the challenges of applying deterministic methods to stochastic settings.

**Experimental Parameters:** Across all tasks, we use a reward discount factor of $\gamma = 0.99$ and planning horizon of $T = 150$ steps. The MCTS search depth is set to $D = 100$ with $n = 100$ simulations per state. In discretized-UCT, actions are discretized into 10 uniformly sampled values. For HOOT and `Stochastic-Power-HOOT`, given action space dimension $m$, we set $\rho = \frac{1}{4^m}$ and $\nu_1 = 4m$. In `Stochastic-Power-HOOT`, we configure HOO tree depth limit to $\bar{H} = 10$ with parameters $b = 5$, $\beta = 20$, and $\alpha = 10$. All algorithms use rollout policy $\pi_0$ initialized as $\widehat{V}(s) = 0$ for all states $s \in \mathcal{S}$.

## H.1. Progressive Widening Parameter Selection

We tune environment-specific PW parameters ($\alpha$, k, c) following:

- Acrobot: $\alpha$=0.6, k=2.5, c=1.2 (timeseries strategy)

- MountainCar: $\alpha$=0.55, k=2.2, c=1.3 (momentum strategy)

- CartPole-IG: $\alpha$=0.5, k=2.0, c=1.0 (adaptive strategy)

- Pendulum: $\alpha$=0.45, k=2.5, c=1.1 (optimistic strategy)

## H.2. Practical Branching Factor Analysis

Our empirical analysis shows that despite theoretical maximum branching factors of $2^D$, practical branching factors remain efficient:

- Average branching factors: 1.86-2.17 across different power values

- More than $95\%$ of nodes have only one child

- Branching concentrates at key decision points

- Power mean backpropagation causes algorithm to focus on promising regions

This demonstrates effective exploration-exploitation balance, with the algorithm thoroughly exploring promising regions while limiting resources on less promising areas.

