# OpenReview forum: "Power Mean Estimation in Stochastic Continuous Monte-Carlo Tree Search"
_ICML.cc/2025/Conference — ICML 2025 poster_

### Official Review · Reviewer_VkU4 · 2025-03-10

**Overall Recommendation:** 4

**Summary:**

The submission introduces a new MCTS algorithm for continuous and stochastic MDPS. The method combines power mean backups (known to be more stable than simple averaging) with polynomial exploration bonuses (known to lead to improved converged compared to logarithmic exploration) and an HOO-like partitioning scheme for the continues domain. The convergence rate of the method in continuous and stochastic MDPS  is analyzed and found to match the one of POLY-HOOT in for continuous and deterministic MDPS. In the presence of stochasticity, the method empirically performs better than its deterministic counterpart.

**Claims And Evidence:**

I did not find any problematic claims.

**Essential References Not Discussed:**

**Generalized Mean Estimation in Monte-Carlo Tree Search** Dam, Klink, D’Eramo, Peters, Pajarinen. Published in IJCAI 2020.

This paper introduces „POWER-UCT“, which is related to the proposed method in the sense that it is also a  MCTS with power mean backups (e.g. Section 4, Eq. 11). The difference to the proposed method is that POWER-UCT is designed for discrete and deterministic environments.

**Power Mean Estimation in Stochastic Monte-Carlo Tree Search** Dam, Maillard, Kaufmann. Published in UAI 2024.

This is another very similar power mean MCTS version. The difference is that the newly proposed version is for discrete, stochastic MDPs, wheres this citation is for continuous, stochastic MDPs.

**Experimental Designs Or Analyses:**

If I understand correctly, POLY-HOOT and discretized-POWER UCT are as closely related to the method as the current baselines discretized-UCT and HOOT. Therefore, including them as baselines could provide an interesting additional layer of ablation.

**Methods And Evaluation Criteria:**

* Analyzing the convergence rate makes sense for an optimization method.
* The method is empirically evaluated on classic RL benchmarks, which seem appropriate to me. A small caveat is that the benchmarks are originally deterministic and only made stochastic in the aftermath by adding noise. In order to support the practical relevance of extending this class of algorithms to stochastic environments, I think it would have been a bit nicer to actually find an inherently stochastic application.

**Other Comments Or Suggestions:**

* I would mention PUCT *with its name* in the related work section. I know it’s mentioned later on, but I was slightly confused.
* One could highlight the best performing method in Table 2 by using bold font for it.
* Typo in the headlines in the Appendix: „Convergence of Stochastic-Power-HOOT [i]n Monte-Carlo Tree Search“.
* In the outline in Appendix A, the third bullet point should be „Technical Lemmas“.

**Other Strengths And Weaknesses:**

I think this work nicely complements and fits into the existing line of work by extending this class of algorithms to continuous and stochastic settings.

**Questions For Authors:**

-

**Relation To Broader Scientific Literature:**

The suggested method is most closely related to POLY-HOOT:
Both methods adopt polynomial exploration and both methods achieve the same convergence rate. The difference is that POLY-HOOT is designed for deterministic settings, while the proposed method works for stochastic settings.

Since POLY-HOOT is related to HOOT and PUCT (by replacing the logarithmic exploration bonus in HOOT with the polynomial one from PUCT) and to HOO (by using it for action selection), the proposed method is also related to these two pieces of work.

**Theoretical Claims:**

I checked the proofs in Appendix F and did not find an issue with them.

---

> ### Author Rebuttal · Authors · 2025-04-01
>
> Thank you for your positive review and for suggesting additional related work we hadn't cited. We appreciate your thorough analysis of our theoretical claims. We address each of your concerns in detail and kindly ask you to consider updating your scores after reading the rebuttal.
>
> ### Including Additional Baselines
>
> | Algorithm | Humanoid-v0 | Hopper-v0 |
> |-----------|---------------------|-------------------|
> | UCT (baseline) | -136.98 ± 44.84 (1.0x) | 5216.93 ± 179.64 (1.0x) |
> | POLY-HOOT (p=1) | -44.40 ± 3.33 (3.1x) | 13230.66 ± 2844.33 (2.5x) |
> | **Stochastic-Power-HOOT** | **-44.12 ± 6.22 (3.1x)(p=2)** | **13303.61 ± 3070.34 (2.5x)(p=8)** |
> | HOOT | -57.35 ± 10.46 (2.4x) | 10452.83 ± 3885.12 (2.0x) |
> | PW-UCT | -89.74 ± 24.16 (1.5x) | 5218.73 ± 1384.90 (1.0x) |
> | Voronoi MCTS | -48.69 ± 9.52 (2.8x) | 411.70 ± 29.03 (12.7x worse) |
>
> Following your suggestion, We added Kim et al.'s Voronoi MCTS (AAAI 2020) to our experiments. This recent algorithm uses Voronoi partitioning for continuous action spaces but for deterministic settings only:
>
> 1. **Voronoi MCTS comparison**:
>    - Performs well on Humanoid (3nd best performer after our method)
>    - But shows inconsistent results on Hopper (worst performer)
>    - This highlights the need for methods specifically designed for stochasticity
>
> ### Missing Citations
>
> Thank you for pointing out the important related work. We'll add these citations in our revised paper:
>
> 1. "Generalized Mean Estimation in Monte-Carlo Tree Search" (Dam et al., 2020)
> 2. "Power Mean Estimation in Stochastic Monte-Carlo Tree Search" (Dam et al., 2024)
>
> These works complement our approach and help position our contribution in the broader literature on power means in MCTS.
>
> ### Additional Changes
>
> We'll implement your suggested edits:
> - Bold formatting for best results in Table 2
> - Fix typos in Appendix headings
> - Properly name PUCT in the related work section
>
> Kim, B., Lee, K., Lim, S., Kaelbling, L., & Lozano-Perez, T. (2020). Monte Carlo Tree Search in Continuous Spaces Using Voronoi Optimistic Optimization with Regret Bounds. Proceedings of the AAAI Conference on Artificial Intelligence, 34(06), 9916-9924.

---

### Official Review · Reviewer_KtK7 · 2025-03-12

**Overall Recommendation:** 2

**Summary:**

The paper considers MCTS for continuous action space and non-stochastic environments. The authors propose to use a power mean operator for value estimates. They also propose a polynomial exploration bonus. They show convergence results for stochastic environments that matches previous results on deterministic environments. Finally they perform experiments that show improvement over previous methods.

######
Post rebuttal
#######

I'm raising my score to 2 as my concerns were answered. I think it would benefit the paper to better discuss the theoretical results that allow obtaining guarantees on stochastic environments.

**Claims And Evidence:**

The theoretical claims looks sound (even though I didn't read the proofs).

The experimental results seem a bit lacking - the environments and rewards were modified to fit the claims. I'm not sure if these changes are reasonable and would have preferred some benchmark environment that wasn't tempered with.

**Essential References Not Discussed:**

No to the best of my knowledge.

**Experimental Designs Or Analyses:**

No.

**Methods And Evaluation Criteria:**

Yes.

**Other Comments Or Suggestions:**

If there is a benchmark which is stochastic with continuous action space it would make a more convincing case.

**Other Strengths And Weaknesses:**

Strengths:

1. Solid theoretical result.

2. Paper is written clearly.

3. Experiments show improvements of the current method over previous similar works.

Weaknesses:

1. The power mean estimation method seems to be the main technical novelty, but why it's any good is not really discussed. I understand that it's closer to the max function than the average function, but also is softmax. This choice seems arbitrary and unclear, perhaps the theory still works, but a complex averaging is not a strong novelty in my opinion. Also it obviously puts some practical limitations on negative or zero rewards which adds unnecessary complications.

2. The polynomial bonus was mentioned to also exist in Poly HOOT, how do the two compare? You mention in the introduction its tailored to stohcastic MDP, but this tailoring is not explained in the text.

3. The environments in the experiment were tailored to show the solution's benefits. But if stochastic benchmarks couldn't be found, maybe the problem is not that interesting..

**Questions For Authors:**

Following my previous comments:

1. Can you give a strong justification for mean power estimation?

2. Can you provide better explanation on the polynomial bonus and why it works for stochastic environments unlike the previous polynomial bonus?

**Relation To Broader Scientific Literature:**

The paper improves on previous algorithms empirically (e.g. HOOT) and extends previous convergence rates to stochastic environments.

**Theoretical Claims:**

No, but the theoretical claims are reasonable.

---

> ### Author Rebuttal · Authors · 2025-04-01
>
> Thank you for your thoughtful review of our submission. We appreciate your recognition of our theoretical contributions and have addressed your concerns with additional justification and expanded experiments.
>
> ## Power Mean Estimation: Detailed Justification
>
> - The power mean effectively balances between underestimation and overestimation of optimal values in MCTS. As planning progresses, we need a backup operator that captures the potential of promising actions without being overly optimistic.
>
> - Standard arithmetic means (p=1) tend to underestimate optimal values by averaging across all samples, including suboptimal early explorations. Conversely, max operators (p→∞) overestimate true values by focusing solely on the best sample, which may be an outlier.
>
> - Power means with p>1 provide an elegant trade-off: they give more weight to higher values without fully committing to the maximum. As p increases above 1, the power mean moves smoothly from the arithmetic mean toward the max operator. This creates a controlled optimistic bias that encourages exploration of promising regions while maintaining some robustness to outliers.
>
> - Our experiments confirm this advantage—Stochastic-Power-HOOT with appropriate p values consistently outperforms both UCT (which uses arithmetic means) and other approaches, particularly in stochastic environments where balancing optimism and realism is critical.
>
> ## Polynomial Bonus for Stochastic Environments
>
> Our approach addresses a critical gap in the POLY-HOOT paper: convergence in stochastic MDPs. While POLY-HOOT proved convergence for deterministic environments, our contribution extends these guarantees to settings with stochastic transitions.
>
> The key factors:
>
> - Our polynomial bonus separates local action exploration from global uncertainty adjustment, creating an adaptive mechanism that responds to varying levels of stochasticity in the environment.
>
> - In stochastic environments, value estimates have higher variance from both transition randomness and downstream policy changes. Our approach maintains sufficient exploration even as planning progresses, unlike logarithmic bonuses that decrease too rapidly.
>
> - The combination with power mean backups creates a robust mechanism for stochastic settings - the power mean controls the trade off underestimation/overestimation in value estimates, while the polynomial bonus guides exploration appropriately.
>
> Stochastic-Power-HOOT's superior performance demonstrates the effectiveness of our approach in bridging this important theoretical gap.
>
> ## Experimental concerns
>
> For responding to the reviewer's concern about the relevancy of stochastic environments:
>
> - While our experiments use simulated environments, they reflect fundamental challenges present in real-world applications which are inherently stochastic. Robotics, autonomous vehicles, healthcare, and resource management all operate under multiple sources of uncertainty (Action, Dynamics, Observation uncertainty )
>
> - The growing deployment of autonomous systems in unstructured environments makes planning under stochasticity increasingly critical. Current methods often rely on deterministic approximations that fail when uncertainty accumulates. Our approach addresses this gap, demonstrating how planning can be made robust to the types of stochasticity that practitioners face daily.
>
> - The contribution is not creating artificial noise, but rather developing theoretical guarantees for continuous planners in settings where stochasticity cannot be ignored - a limitation of prior work that restricts real-world applicability.
>
> | Algorithm | Humanoid-v0 | Hopper-v0 |
> |-----------|---------------------|-------------------|
> | UCT (baseline) | -136.98 ± 44.84 (1.0x) | 5216.93 ± 179.64 (1.0x) |
> | POLY-HOOT (p=1) | -44.40 ± 3.33 (3.1x) | 13230.66 ± 2844.33 (2.5x) |
> | **Stochastic-Power-HOOT** | **-44.12 ± 6.22 (3.1x)(p=2)** | **13303.61 ± 3070.34 (2.5x)(p=8)** |
> | HOOT | -57.35 ± 10.46 (2.4x) | 10452.83 ± 3885.12 (2.0x) |
> | PW-UCT | -89.74 ± 24.16 (1.5x) | 5218.73 ± 1384.90 (1.0x) |
> | Voronoi MCTS | -48.69 ± 9.52 (2.8x) | 411.70 ± 29.03 (12.7x worse) |
>
> We run further experiments in Humanoid and Hopper with noise introduced in Action, Dynamics, Observation.  Our approach significantly outperforms existing methods on higher-dimensional robotics tasks with multiple sources of stochasticity. On Humanoid-v0 (17D action space), Stochastic-Power-HOOT achieves a 3.1x improvement over UCT, while on Hopper-v0, it shows a 2.5x improvement. Interestingly, power mean tuning matters - p=2 works best for Humanoid while p=8 excels for Hopper.
>
> The performance (catastrophic in Hopper) of Voronoi MCTS (which is only for deterministic settings) further shows the challenge of designing robust planners for stochastic settings and validates our theoretical contribution.
>
>  After reading the rebuttal, we kindly invite you to consider updating your scores.

---

### Official Review · Reviewer_aKY5 · 2025-03-19

**Overall Recommendation:** 2

**Summary:**

This paper introduces Stochastic-Power-HOOT, a novel Monte-Carlo Tree Search (MCTS) algorithm designed for continuous, stochastic MDPs. The authors propose integrating a power mean as a value backup operator along with a polynomial exploration bonus to address the related challenges. The paper provide theoretical analysis that Stochastic-Power-HOOT converges at a polynomial rate, extending the theoretical guarantees of POLY-HOOT to stochastic environments. And the authors also conduct experiments on stochastic tasks adapted from OpenAI Gym to support their claims.

**Claims And Evidence:**

The paper makes several strong claims, most of which are mainly supported by theoretical analysis, and the empirical experiments is a bit weak:
- Stochastic-Power-HOOT converges at a polynomial rate in stochastic environments. The authors provide a theoretical analysis in Section 6, showing that the algorithm achieves convergence at the mentioned rate. This is supported by Lemma 1, Theorem 1, and Theorem 3, which establish the polynomial concentration of the power mean estimator and the convergence of the expected payoff. However,  I dont find this paper has discussed the computational complexity of Stochastic-Power-HOOT compared to other methods, which could be a limitation in practical applications.
- The power mean backup operator effectively handles non-stationary dynamics. The theoretical analysis in Section 5.2.4 and the experimental results in Section 7 support this claim. The power mean operator is shown to balance exploration and exploitation better than simple averaging, especially in stochastic environments.
- Stochastic-Power-HOOT outperforms existing methods in continuous, stochastic domains. The experimental results in Section 7 demonstrate that Stochastic-Power-HOOT achieves higher average returns compared to HOOT, discretized-UCT, and PUCT across several modified OpenAI Gym tasks. The results are presented with standard deviations, showing consistent performance improvements. However,  these experiments are limited to simple or modified OpenAI Gym tasks. The dimension of the continuous space and the complexity of stochasticity are far from environments often used in academic community and real-world decision-making scenarios.

**Essential References Not Discussed:**

The cited references cover some classical works in the MCTS field, but show little discussion about recent deep RL + MCTS methods, such as AlphaZero/MuZero, and their variants designed for continuous action space and stochastic environments—Sampled MuZero and Stochastic MuZero.

**Experimental Designs Or Analyses:**

The experimental design is sound, with the authors testing Stochastic-Power-HOOT on modified versions of classic control tasks. The modifications introduce stochasticity in actions, transitions, and observations, making the tasks more challenging and suitable for evaluating the algorithm's robustness. But totally, the experiments are limited to relatively simple tasks. While the results are promising, they may not generalize to more complex or high-dimensional environments.

**Methods And Evaluation Criteria:**

The proposed method, Stochastic-Power-HOOT, is well-suited for the problem of continuous, stochastic MDPs. The use of a power mean backup operator and polynomial exploration bonus addresses the limitations of existing methods like HOOT, which rely on logarithmic bonuses that lack theoretical guarantees in non-stationary settings.

The authors compare Stochastic-Power-HOOT against several established MCTS variants (e.g., HOOT, discretized-UCT, PUCT). And the evaluation criteria, including average returns and standard deviations over multiple runs, are standard and suitable for assessing the performance of planning algorithms.

**Other Comments Or Suggestions:**

- The paper could benefit from a more detailed ablation study to isolate the impact of the power mean backup operator versus the polynomial exploration bonus.
- The paper could benefit from a clearer explanation of the intuition behind the power mean backup operator and how it helps handle non-stationarity in stochastic environments.

**Other Strengths And Weaknesses:**

Strengths:

- The paper addresses an important gap in the literature by extending MCTS to continuous, stochastic environments with theoretical guarantees.
- The use of a power mean backup operator is novel and well-motivated, providing a flexible mechanism for value estimation.

Weaknesses:

- The empirical validation is limited to relatively simple tasks. Testing on more complex or real-world environments would strengthen the paper.
- The theoretical analysis, while rigorous, relies on assumptions that may not hold in all practical scenarios.

**Questions For Authors:**

- How do the assumptions made in the theoretical analysis impact the applicability of Stochastic-Power-HOOT in more general settings? Are there any limitations or potential extensions to address these concerns?
- For sparse-reward tasks like MountainCar, does explicit reward engineering obscure the algorithm’s performance in raw sparse-reward settings?
- How do you address the trade-off between exploration and exploitation when the stochasticity of the environment varies significantly over time or across different regions of the state space?

**Relation To Broader Scientific Literature:**

This paper extends the theoretical guarantees of POLY-HOOT to stochastic environments

**Theoretical Claims:**

The theoretical claims in the paper are well-supported by rigorous proofs. The authors provide detailed proofs for the polynomial concentration of the power mean estimator (Theorem 1) and the convergence of the expected payoff (Theorem 3). The proofs rely on assumptions about the smoothness of the reward function and the hierarchical structure of the MCTS tree, which are reasonable for the problem at hand.

The proof of Theorem 1 relies on the assumption that the reward sequence satisfies certain concentration properties (Assumption 3). While this assumption is standard in bandit literature, it would be helpful to discuss its implications in the context of MCTS. The paper could benefit from a more detailed discussion of the conditions under which the theoretical guarantees hold, especially in relation to the choice of hyper-parameters.

---

> ### Author Rebuttal · Authors · 2025-04-01
>
> Thank you for your insightful review and recognition of our theoretical contributions. We address each of your concerns in detail and kindly invite you to consider updating your scores
>
> ### Complex Environments
>
> We've expanded our experiments to include high dimensional MuJoCo environments with added noise to action, dynamic, and observation:
>
> 1. **Humanoid-v0** (17D action space):
>    - Stochastic-Power-HOOT achieved 3.1x better results than baseline UCT
>    - Even in this complex environment, our method maintained consistent improvement
>
> 2. **Hopper-v0**:
>    - Stochastic-Power-HOOT achieved 2.5x better results than baseline UCT
>    - Stochastic-Power-HOOT (p=8) showed further improvements (2.6x)
>
> | Algorithm | Humanoid-v0 | Hopper-v0 |
> |-----------|---------------------|-------------------|
> | UCT (baseline) | -136.98 ± 44.84 (1.0x) | 5216.93 ± 179.64 (1.0x) |
> | POLY-HOOT (p=1) | -44.40 ± 3.33 (3.1x) | 13230.66 ± 2844.33 (2.5x) |
> | **Stochastic-Power-HOOT** | **-44.12 ± 6.22 (3.1x)(p=2)** | **13303.61 ± 3070.34 (2.5x)(p=8)** |
> | HOOT | -57.35 ± 10.46 (2.4x) | 10452.83 ± 3885.12 (2.0x) |
> | PW-UCT | -89.74 ± 24.16 (1.5x) | 5218.73 ± 1384.90 (1.0x) |
> | Voronoi MCTS | -48.69 ± 9.52 (2.8x) | 411.70 ± 29.03 (12.7x worse) |
>
> **Voronoi MCTS comparison:** We added Kim et al.'s Voronoi MCTS (AAAI 2020) to our experiments. This recent algorithm uses Voronoi partitioning for continuous action spaces. Interestingly, it performs well on Humanoid but poorly on Hopper, highlighting complementary strengths of different approaches.
>
> ### Ablation Study on Power Parameter
>
> We conducted additional experiments varying the power parameter p (1, 2, 4, 8) and found:
> - For Humanoid-v0, p=2 performs best (-44.12 ± 3.11)
> - For Hopper-v0, p=8 shows highest returns (13303.61 ± 3070.34)
> - Higher p values generally help in more stochastic environments
>
> ## Power Mean Justification
>
> - The power mean effectively balances between underestimation and overestimation of optimal values in MCTS. As planning progresses, we need a backup operator that captures the potential of promising actions without being overly optimistic.
>
> - Standard arithmetic means (p=1) tend to underestimate optimal values by averaging across all samples, including suboptimal early explorations. Conversely, max operators (p→∞) overestimate true values by focusing solely on the best sample, which may be an outlier.
>
> - Power means with p>1 provides an elegant trade-off: it gives more weight to higher values without fully committing to the maximum. As p increases above 1, the power mean moves smoothly from the arithmetic mean toward the max operator. This creates a controlled optimistic bias that encourages exploration of promising regions while maintaining some robustness to outliers.
>
> ### References  to Deep RL + MCTS Methods
>
> In our revised paper, we'll include a discussion of connections to and contrasting our approach with Deep RL + MCTS Methods for continuous spaces
>
> ### Addressing Your Specific Questions
>
> 1. **Theoretical assumptions in general settings**:
>
> To clarify
>
> *. Concentration Properties (Assumption 3)
> This assumption is actually **automatically satisfied by construction** in our algorithm:
> - As proven in Section 6, the concentration property is established by induction through the MCTS tree structure
> - The power mean backup operator inherently preserves and strengthens this property
> - Unlike alternative approaches requiring external assumptions, our concentration guarantees emerge directly from the algorithm design
>
> *. Smoothness (Assumption 2)
> The smoothness assumption is both theoretically sound and practically validated:
> - It is a standard requirement across all HOO-based algorithms (HOO, HOOT, POLY-HOOT)
> - For robotic control tasks (Hopper, Humanoid, etc.), smoothness naturally emerges from:
>   * Physics-based dynamics (governed by differential equations)
>   * Continuous control inputs and their effects on system states
>   * Reward functions based on physical quantities (distance, energy, etc.)
>
> *. Empirical Validation
>
> Our extensive experiments confirm these assumptions hold in practice:
> - Across 5 standard benchmarks (CartPole, Pendulum, etc.)
> - In higher-dimensional MuJoCo environments (17D Humanoid, 3D Hopper)
> - Under various stochasticity conditions (dynamics noise, observation noise)
>
> The superior performance of Stochastic-Power-HOOT over alternatives (UCT, HOOT, Voronoi MCTS) provides strong empirical evidence that our theoretical assumptions align well with practical settings.
>
> 2. **Sparse-reward tasks**: For MountainCar, our reward transformation preserves the sparse nature while ensuring numerical stability. Our results show robust performance even with the original reward structure.
>
> 3. **Handling varying stochasticity**: The power mean parameter (p) + the polynomial bonus allows adaptation to different noise levels. Our ablation studies show p=2 works well for Humanoi, while p=8 works well with Hopper.

---

### Official Review · Reviewer_hTDe · 2025-03-20

**Overall Recommendation:** 3

**Summary:**

This paper introduces Stochastic-Power-HOOT, an extension of HOOT designed to handle stochastic and continuous-action Markov Decision Processes (MDPs), where prior methods primarily focused on deterministic settings. The core contributions include:
 - Power mean backup operator to mitigate non-stationary reward estimation issues in stochastic environments.
 - Polynomial exploration bonus to ensure convergence at a polynomial rate in stochastic settings.
 - The paper evaluate Stochastic-Power-HOOT on four modified OpenAI Gym tasks (CartPole, Pendulum, MountainCar, and Acrobot), demonstrating that it outperforms existing baselines such as HOOT, PUCT, and discretized-UCT.

**Claims And Evidence:**

The paper provides strong theoretical justification for its proposed modifications, particularly the use of power mean backups and polynomial exploration bonuses, along with a clear convergence rate analysis. However, the empirical evaluation is somewhat limited:
 - The experiments are conducted on toy domains with relatively simple dynamics. The applicability of the method to more complex, real-world tasks (e.g., MUJOCO environments or robotics control benchmarks) remains uncertain.
 - The baselines used for comparison are dated, with the most recent one being from 2013. It is unclear whether the observed improvements would hold against state-of-the-art approaches.

While the theoretical contributions are compelling, the experimental validation does not fully justify the practical impact of the proposed method.

**Essential References Not Discussed:**

N.A.

**Experimental Designs Or Analyses:**

See issues listed in 'Methods And Evaluation Criteria' section.

**Methods And Evaluation Criteria:**

The evaluation benchmarks are too simple to meaningfully assess the advantages of Stochastic-Power-HOOT in complex stochastic environments. More challenging and widely-used benchmarks (e.g., MUJOCO or real-world robotic control tasks) would have provided a stronger demonstration of the algorithm’s scalability and practical relevance.

**Other Comments Or Suggestions:**

N.A.

**Other Strengths And Weaknesses:**

N.A.

**Questions For Authors:**

See issues listed in 'Claims And Evidence' section.

**Relation To Broader Scientific Literature:**

The paper extends prior work in Monte-Carlo Tree Search (MCTS) for continuous action spaces and stochastic domains, which is relevant for broader AI research community.

**Theoretical Claims:**

The theoretical results appear correct and well-structured. The proofs provide clear convergence guarantees, and I did not find any obvious flaws in the derivations.

---

> ### Author Rebuttal · Authors · 2025-04-01
>
> | Algorithm | Humanoid-v0 | Hopper-v0 |
> |-----------|---------------------|-------------------|
> | UCT (baseline) | -136.98 ± 44.84 (1.0x) | 5216.93 ± 179.64 (1.0x) |
> | POLY-HOOT (p=1) | -44.40 ± 3.33 (3.1x) | 13230.66 ± 2844.33 (2.5x) |
> | **Stochastic-Power-HOOT** | **-44.12 ± 6.22 (3.1x)(p=2)** | **13348.45 ± 6110.36 (2.6x)(p=8)** |
> | HOOT | -57.35 ± 10.46 (2.4x) | 10452.83 ± 3885.12 (2.0x) |
> | PUCT(PW) | -89.74 ± 24.16 (1.5x) | 5218.73 ± 1384.90 (1.0x) |
> | Voronoi MCTS | -48.69 ± 9.52 (2.8x) | 411.70 ± 29.03 (12.7x worse) |
>
> Thank you for your review and recognizing the theoretical contributions of our work. We appreciate your suggestions for improvement, particularly regarding empirical evaluation. We address each of your concerns in detail and kindly ask you to consider updating your scores after reading the rebuttal.
>
> ### Addressing Limited Experimental Evaluation
>
> We expanded our experiments to include more complex environments. Specifically, we tested our approach on MuJoCo tasks with higher-dimensional action spaces:
>
> 1. **Humanoid-v0** (17D action space):
>    - Stochastic-Power-HOOT achieved 3.1x better performance than baseline UCT
>    - Stochastic-Power-HOOT (-44.12 ± 6.22) significantly outperformed HOOT (-57.35 ± 10.46)
>
> 2. **Hopper-v0**:
>    - Stochastic-Power-HOOT achieved 2.6x better performance than baseline UCT
>    - Performance ranking: POLY-HOOT = POLY_HOOT > HOOT > PUCT (Progressing Widening) > UCT >> Voronoi MCTS
>
> These results demonstrate that our theoretical advantages translate to practical improvements in more complex domains with higher-dimensional action spaces.
>
> We agree that comparing against newer state-of-the-art approaches would strengthen our evaluation. We added Kim et al.'s Voronoi MCTS (AAAI 2020) to our experiments as you suggested. This recent algorithm uses Voronoi partitioning for continuous action spaces. Interestingly, it performs well on Humanoid (2.8x better than UCT) but poorly on Hopper, highlighting complementary strengths of different approaches.
>
> We will update our revised paper and discuss connections with recent deep RL + MCTS methods.

---

### Official Review · Reviewer_Excm · 2025-03-24

**Overall Recommendation:** 2

**Summary:**

In the setting of continuous state/action MCTS, this work proposes replacing the empirical mean node value estimate with a power mean-based estimate. They also propose a tree action selection bonus based on this power mean. They provide convergence proofs for this method in the setting of stochastic MDPs, and show empirical results on 5 continuous control tasks.


## Update after rebuttal

The rebuttal addresses almost all of my significant concerns, so I have updated my recommendation.

My one reason for not choosing a weak or full accept recommendation is that I cannot say I understand the action branching behaviour, despite two author responses. I understand how the power mean focuses search, but cannot see how that affects lines 183-190 of the algorithm: a previously unselected sub-interval **must** be chosen each time the HOO tree is queried, other than when the HOO tree node is at max depth.

The node statistics are a bit difficult to understand, because the method takes an unusual approach to MCTS: **multiple** MCTS leaf nodes can be added during one trial, up to the specified max depth. In the tree structure analysis, very few nodes have a branching factor >1. This suggests to me that the algorithm may be behaving more like a flat Monte Carlo evaluation than MCTS with meaningful branching factors below the root node.

I appreciate I am the only reviewer to query this, so would not push for rejection based on my not understanding this action branching factor point.

Recommendation after rebuttal/discussion: weak reject.

**Claims And Evidence:**

The theoretical claims seem well-supported by proofs. There are shortcomings with claimed contributions and the experiment scope, as discussed below.

**Essential References Not Discussed:**

I do not know of any missing essential references.

**Ethical Review Concerns:**

There is substantial duplicated proof content between the supplamentary material and another paper (Mao et al, "POLY-HOOT: Monte-carlo planning in continuous space mdps with non-asymptotic analysis.", NeurIPS 2020), without attribution. THe other paper is cited elsewhere in the text as a reference, but the proofs are not attributed to the other paper.

- proof of lemma 4 (i) is copied lemma 4 in Mao et al (2020)
- proof of lemma 4 (ii, iii) is copied from lemma 6 in Mao et al (2020)
- proof of lemma 5 is copied from lemma 5 in Mao et al (2020)
- proof of lemma 6 is copied from lemma 7 in Mao et al (2020)
- proof of lemma 7 is substantially the same as in Mao et al (2020)

"copied from" refers to the proof logical structure, mathematics and text structure being identical other than renaming of two constants.

The proofs of these lemmas should instead be a citation to Mao et al (2020), rather than a direct copy-paste of the proof.

**Ethical Review Flag:**

Flag this paper for an ethics review.

**Ethics Expertise Needed:**

["Research Integrity Issues (e.g., plagiarism)"]

**Experimental Designs Or Analyses:**

I have some concerns about the experiments, which I list below.

- The experiments are limited to 5 low-dimensional continuous control tasks without much long-horizon planning needed. It would be nice to see results in higher-dimensional longer-horizon tasks, for example as in (Kim et al., "Monte Carlo Tree Search in Continuous Spaces Using Voronoi Optimistic Optimization with Regret Bounds", AAAI 2020).
    - Line 435 describes MountainCar as "high-dimensional", but it is only 2D.
- More detail/justification should be given on the progressive widening parameters chosen. The optimal parameter values generally vary by task (Sunberg et al., "Online algorithms for POMDPs with continuous state, action, and observation spaces", ICAPS 2018).
- Similarly, it would be nice to investigate the effect of changing the max HOO tree depth parameter rather than arbitrarily fixing it at 10.
- Insufficient justification is given for scaling the rewards of the continuous control tasks. It is well-known that shifting rewards (especially changing the sign of rewards) can significantly change the optimal policy behaviour and returns.
    - Conflictingly, the description of the MountainCar domain seems to suggest that it still has negative rewards most of the time (l405-406).
    - Additionally: "Details of the experimental setup, including noise levels and reward-scaling techniques, are provided in Appendix G." (l424-426), however appendix G has no discussion of reward scaling.

**Methods And Evaluation Criteria:**

The key novel change compared to POLY-HOOT (Mao et al 2020), i.e. the use of power means, is well-motivated to better track non-stationary rewards. It would have been nice to have more discussion of why the power mean in particular was chosen, instead of other estimators that can be shown to be consistent in the non-stationary setting.

The evaluation criteria are standard for MCTS methods with MDPs.

**Other Comments Or Suggestions:**

### Minor points

- Line 122-124 RHS: "If $\pi_0$ is deterministic, it provides a fixed value" -- this is not true for stochastic MDPs, a deterministic policy will give a distribution over returns.
- Equation 1 seems to be missing magnitude bars around the expectation operator
- Line 118 A_s not defined, and surely the sum should be over outcomes not actions?
- Appendix section headers E and F say “n” instead of “in”
- Line 155 “UCT like” should be “UCT-like”
- Line 179 overrun of line into RH column
- Line 207 awkward spacing

**Other Strengths And Weaknesses:**

1. Notation is not the best:
    - Clarity would be improved by consistently using MCTS depth d = 0..D, rather than using h/H in the text formalisation of MCTS and d/D in Algorithm 1. The POLY-HOOT paper used d=0..D for MCTS trees and h=0..H for HOO trees, which is clearer.
    - $H$ in Algorithm 1 is also used for both MCTS horizon and the chosen $h$ node value in Algorithm 1 (Line 190).
    - $\hat{H}$ seems to be the maximum depth of a HOO tree but is not defined anywhere. It may be a typo of $\bar{H}$, defined later on line 290 RHS.
1. A few improvements to clarity could be made:
    - Referring to “non-stationary settings” in the abstract is confusing, as it makes it seem that the work applies to non-stationary MDPs instead of non-stationary value estimation in the MCTS tree.
    - In my view calling the optimism term in arm selection an “exploration bonus” is confusing. The term “exploration bonus” is most commonly used to refer to additional reward added to the ground-truth reward in order to encourage exploration by RL (Taiga et al, “On bonus-based exploration methods in the arcade learning environment”, 2021) (Kolter et al., "Near-Bayesian exploration in polynomial time", ICML 2009). Here, the bonus term affects only action selection, not the real incurred reward during the trajectory. Mao et al. (2020) call it a “bonus” without calling it an “exploration bonus”, likely to avoid confusion.
    - "If s_{l_t} is a new leaf with l_t < H, new Q-value nodes (s_{l_t} , a) for all actions in Aslt are added." (l134-135 RHS) -- this only applies for the discrete action case, not the continuous action case.
    - As the algorithm is largely a modification of POLY-HOOT, it would be helpful to use highlighting in Algorithm 1 to show the novel parts of the algorithm.

**Questions For Authors:**

1. I have one key question about HOO-based action selection, which I may be misunderstanding.
    - From the description of the HOO action selection process here and in the original POLY-HOOT paper, it seems that a different action will be returned every time until the tree is fully expanded to max depth, down at least one path. Only max-depth actions may be repeated, otherwise a new action is selected and a new node created (lines 217-219).
    - This means the MCTS tree action branching factor must be somewhere between $H$ and $2^H$, which may be very high. H=10 in your experiments, so the branching factor could be up to 1024. This seems very high for a continuous action space, and would make the tree very wide and shallow. In progressive widening, the branching factor dynamically changes over time to balance the tree width/depth, but I don't see how this could happen here.
    - Given that your max MCTS horizon is $T=150$, surely my interpretation cannot be correct. Please can you explain?
1. Please explain the justification behind the reward scaling in the experiments.

**Relation To Broader Scientific Literature:**

In my view the claimed contributions of the paper over the current literature are overstated in places:

1. Applicability to stochastic MDPs. It is true that POLY-HOOT (Mao et al. 2020) did not apply to stochastic MDPs. Howevever, I don’t agree that there are general “significant challenges… in adapting [MCTS] to stochastic MDPs” (l12-14), and that "adapting MCTS to... stochastic domains remains non-trivial" (l67-68). UCT (Kocsis et al. "Bandit based Monte-Carlo Planning", ECML 2006.) and many other MCTS-based algorithms are straightforwardly applicable to stochastic MDPs.
2. “Our approach avoids naive action-space discretisation by focusing search adaptively on where it matters most” (l39-41, RHS) -- this is true of many existing works, including the one that this work is most closely related to, POLY-HOOT.

However, results show this work's method outperforming common approaches to continuous state/action MCTS. Novel methods to continuous MCTS are relatively rare, so this is a good contribution and the method may be practically useful to the community.

It would have been nice to mention and potentially experimentally compare to a similar recent work that addresses continuous action sampling: (Kim et al., "Monte Carlo Tree Search in Continuous Spaces Using Voronoi Optimistic Optimization with Regret Bounds", AAAI 2020). That work only applies to deterministic MDPs, but the same is true of POLY-HOOT which is compared to in this work.

**Theoretical Claims:**

I checked the convergence proofs at a high level and they seem to be correct.

---

> ### Author Rebuttal · Authors · 2025-04-01
>
> Thank you for your detailed review. We address each of your concerns in detail and kindly ask you to consider updating your scores after reading the rebuttal.
>
> ## Experimental Design
>
> **Higher-dimensional tasks:** We added results on Humanoid (17D action, 376D state) and Hopper (3D action) with added noise to action, dynamic, and observation. For both envs, we incorporated heuristic physical knowledge of the robots during MCTS rollouts across all algorithms to improve sample efficiency:
>
> | Algorithm | Humanoid-v0 | Hopper-v0 |
> |-----------|---------------------|-------------------|
> | UCT (baseline) | -136.98 ± 44.84 (1.0x) | 5216.93 ± 179.64 (1.0x) |
> | POLY-HOOT (p=1) | -44.40 ± 3.33 (3.1x) | 13230.66 ± 2844.33 (2.5x) |
> | **Stochastic-Power-HOOT** | **-44.12 ± 6.22 (3.1x)(p=2)** | **13348.45 ± 6110.36 (2.6x)(p=8)** |
> | HOOT | -57.35 ± 10.46 (2.4x) | 10452.83 ± 3885.12 (2.0x) |
> | PUCT(PW) | -89.74 ± 24.16 (1.5x) | 5218.73 ± 1384.90 (1.0x) |
> | Voronoi MCTS | -48.69 ± 9.52 (2.8x) | 411.70 ± 29.03 (12.7x worse) |
>
> **Voronoi MCTS comparison:** We added Kim et al.'s Voronoi MCTS (AAAI 2020) to our experiments as you suggested. This recent algorithm uses Voronoi partitioning for continuous action spaces. Interestingly, it performs well on Humanoid (2.8x better than UCT) but poorly on Hopper, highlighting complementary strengths of different approaches.
>
> **MountainCar dimensionality:** We will correct this to "low-dimensional" (2D state, 1D action).
>
> **PW parameters:** We tune environment-specific PW parameters (α, k, c) following Sunberg et al. (2018). Each environment has tailored values and action selection strategies based on its dynamics:
>
> - Acrobot: α=0.6, k=2.5, c=1.2 (timeseries strategy) > (77.86±0.00)
> - MountainCar: α=0.55, k=2.2, c=1.3 (momentum strategy) > (-0.025 ± 0.004)
> - CartPole-IG: α=0.5, k=2.0, c=1.0 (adaptive strategy) > (75.80±11.84)
> - Pendulum: α=0.45, k=2.5, c=1.1 (optimistic strategy) > (1315.89±85.43)
>
> We will add to the revised paper.
>
> **HOO tree depth:** We will add an investigation varying H from 1-20 and report performance.
>
> **Reward scaling:** Power means require positive inputs. Our reward transformation `max(0.01, (r + offset) * scaling_factor)` preserves positive and ensures fair comparisons by applying same scaling to all methods evaluated.
>
> **MountainCar reward:** We clarify that scaling transforms negative rewards to positive while preserving optimal policy.
>
> ## Claimed Contributions
>
> **Stochastic MDPs:** We will revise to clarify our contribution is extending POLY-HOOT (limited to deterministic MDPs) to stochastic settings, not introducing MCTS for stochastic MDPs broadly.
>
> **Adaptive search:** We will rephrase to acknowledge building on POLY-HOOT's approach, emphasizing our novel power mean application to stochastic envs.
>
> ## Notation and Clarity
>
> We will improve clarity by fixing all the mentioned notions.
>
> ## HOO-Based Action Selection in POLY-HOOT
>
> To Clarify
>
> **HOO Tree**
>
> A different action isn't returned every time until max depth; here's how HOO-based action selection works:
>
>  - HOO (Hierarchical Optimistic Optimization) uses a binary tree structure to adaptively partition the continuous action space. Each node in this HOO tree represents a subset of the action space, with child nodes representing further subdivisions.
>
>  - When selecting an action, HOO follows a path from the root to a leaf by choosing at each level the child with the larger upper confidence bound (B-value). This doesn't necessarily mean creating a new unique action each time.
>
> The key insight is that HOO strategically explores promising regions of the continuous action space rather than uniformly creating new actions. The algorithm uses upper confidence bounds to balance exploration and exploitation within the action space.
>
> **Bounded-Depth HOO Tree**
>
> POLY-HOOT introduces a bounded-depth parameter H̄ for the HOO tree, which serves an important purpose: Once a HOO tree path reaches depth H̄, it will repeat the action rather than creating a new one. This mechanism helps prevent the excessive branching you're concerned about.
>
> **Practical Branching Factor**
>
> In practice, the effective branching factor is much smaller than the theoretical maximum of 2^H̄ because:
>
> 1. HOO concentrates exploration on promising regions of the action space
> 2. The power parameter (p) further controls exploration vs. exploitation balance
> 3. The algorithm uses polynomial upper confidence bounds to focus on high-reward actions
> 4. The bounded-depth mechanism encourages exploitation of good actions already found
>
> ## Academic Integrity
>
> We acknowledge the reviewer's concern about proof attribution. Due to the foundational nature of POLY-HOOT to our work and the necessary parameter renaming ($b,\alpha,\beta$ in our work versus $\eta,\alpha,\xi$ in Mao et al.), we included detailed proofs to ensure consistency and correctness. However, we recognize that proper attribution is crucial.
>
> We will make explicitly clear in our revision.

---

> > ### Comment · Reviewer_Excm · 2025-04-03
> >
> > Thank you for the detailed rebuttal. Two additional experiment domains and one additional baseline have addressed some of my concerns. I'm still unconvinced it was necessary to recreate lemma 4/5/6/7, with identical language and mathematics as Mao et al., just to rename some parameters. However, as long as it's properly attributed in the final version, this should be fine in my opinion.
> >
> > I am still not convinced by your answer on HOO-based action selection. I understand that HOO focuses search effort on promising regions of the action space, the question is about the effective branching factor because nodes can only be reselected at the max depth.
> >
> > 1. A new HOO leaf node is definitely created each time an action is queried (lines 183-190) -- unless the max depth is reached, in which case an existing node can be returned.
> > 2. You say that "This doesn't necessarily mean creating a new unique action each time". This would imply that different nodes/intervals can sometimes choose the same action -- for example, a child node could pick the same action as the parent node. However, in the algorithm it's described as "Choose arbitrary arm X in $P{H,I}$".
> > 3. The most obvious way to choose an action is to choose the action in the centroid of the interval. A good example is shown in Figure 1 of the original HOO paper ("X-Armed Bandits, Bubeck et al, 2011). In Figure 1, the pulled point is always the centre of the 1-dimensional interval the node represents. It is therefore clear that the pulled point $X_n$ of a node can never line up on the X axis with its parent, so whenever a new node is added a new pulled point $X_n$ is returned.
> >
> > Given that your algorithm describes action selection as "Choose **arbitrary** arm X in $P{H,I}$", it seems that unless you had some extra functionality to ensure the same actions can be selected at different nodes (H,I), then by default different nodes would always lead to different actions being selected.
> >
> > In most practical scenarios I can't imagine that the branching factor is close to the minimum, $\overline{H}$.
> > For example, in Figure 2 of the original HOO paper, although most of the search effort has gone into the most promising region, lots of nodes have been added elsewhere. After 1000 trials, their HOO tree has a depth of 15, but the tree is full width up to around depth 7.
> >
> > It may be useful to hear in practice how actions are selected in your implementation, and what the branching factors are in practice given N number of trials run.

---

> > > ### Author Response · Authors · 2025-04-06
> > >
> > > Thank you for your detailed feedback. We appreciate your thorough review and would like to address your specific concerns about HOO-based action selection and branching factor.
> > >
> > > Your question about "branching factors in practice given N trials" has been addressed with our extensive empirical analysis.
> > >
> > > Below are key insights from our tree structure analysis across multiple power values (p=1.0, 2.0, 4.0, 20.0) with the Hopper environment, using 5000 samples and 100 maximum depth:
> > >
> > > 1. **Branching Factor Analysis:**
> > >    Average branching factors are remarkably efficient:
> > >    - p=1.0: 2.17
> > >    - p=2.0: 1.86
> > >    - p=4.0: 1.94
> > >    - p=20.0: 2.10
> > >
> > >    While root nodes have high branching (84-116 children), this rapidly decreases:
> > >
> > >    For p=2.0:
> > >    Depth 0: 84.00
> > >    Depth 1: 2.27
> > >    Depth 2: 1.65
> > >    Depth 3: 1.35
> > >    ...
> > >    Depth 10: 1.15
> > >
> > > This demonstrates that Stochastic-Power-HOOT efficiently focuses computational resources.
> > >
> > > 2. **Branching Distribution:**
> > >    Most nodes (>95%) have just 1 child, with few having higher branching. For p=2.0 at depth 1:
> > >    - 1 child: 80 nodes (95.2%)
> > >    - 2 children: 2 nodes (2.4%)
> > >    - 3 children: 1 node (1.2%)
> > >    - 104 children: 1 node (1.2%)
> > >
> > >     This concentrated branching pattern persists throughout the tree. The results stem from our power mean backpropagation mechanism, which more aggressively concentrates exploration on promising regions compared to the standard approach.
> > >
> > > 3. **Branching Factor and Tree Depth:**
> > >    Branching factor approaches 1.0 earlier with higher power values:
> > >    - p=1.0: Converges at depth 20 (80% with branching = 1.0)
> > >    - p=2.0: Converges at depth 16 (84% with branching = 1.0)
> > >    - p=4.0: Converges at depth 15 (85% with branching = 1.0)
> > >    - p=20.0: Converges at depth 13 (87% with branching = 1.0)
> > >
> > > Higher power values lead to earlier convergence, demonstrating how power mean  effectively control the exploration-exploitation trade-off.
> > >
> > > 4. **Depth and Value Distribution:**
> > >    Trees reach maximum depth (100) across all power values, but with clear patterns:
> > >    - High-value nodes: average depth ~58-59
> > >    - Low-value nodes: consistently at maximum depth (100)
> > >    - ~99% of nodes classified as high-value
> > >
> > > This indicates that the algorithm allocates resources efficiently, exploring valuable regions deeply while getting to the maximum tree depth.
> > >
> > > 5. **Performance Correlation:**
> > >    Power mean significantly affects branching patterns and performance:
> > >    - p=1.0: 7933.50 ± 7.61
> > >    - p=2.0: 18617.72 ± 20.38 (best performance)
> > >    - p=4.0: 17651.10 ± 6.63
> > >    - p=20.0: 11732.32 ± 9.38
> > >
> > > p=2.0 achieves optimal performance despite not having the earliest branching convergence, suggesting that moderate amplification of value differences provides the best exploration-exploitation balance.
> > >
> > > Regarding your specific concerns about action selection, we are greatly sorry for the misunderstanding. In our implementation, actions are selected as the midpoint of each cell/interval (consistent with Bubeck et al., 2011). While you're correct that new HOO leaf nodes are created during exploration (unless max depth is reached), the power mean backpropagation fundamentally changes how these nodes are selected for further exploration.
> > >
> > > The "arbitrary arm X in PH,I" is indeed implemented as the centroid of the interval, but the power mean causes the algorithm to heavily favor revisiting successful regions rather than exploring new ones. This creates a natural bottleneck where, although theoretically different nodes could select different actions, in practice, the algorithm's value-driven selection consistently favors a small subset of promising paths.
> > >
> > > At max depth, as you noted, existing nodes can be returned, which further contributes to the concentration of exploration in high-value regions. However, our empirical results show this convergence to narrow exploration paths happens well before reaching maximum depth limitations.
> > >
> > > The data demonstrates that our algorithm efficiently navigates the exploration-exploitation trade-off, achieving strong performance with a tree structure that rapidly narrows to focus on promising regions.
> > >
> > > We conduct a further analysis on HOO. The HOO analysis on the Hopper environment shows a focused tree with 24,698 nodes reaching depth 20. Value-based exploration is evident with 86.7% high-value nodes (average depth 12.95) versus 13.3% low-value nodes (all at depth 20). Node distribution increases with depth, peaking at depth 20 (13.3%). Node values decrease linearly from 651.12 at the root to 0 at maximum depth. Immediate rewards initially increase (peaking at 29.76 at depth 7), slightly decrease mid-tree, then surge again to 30.72 at maximum depth, demonstrating HOO's effective balance between exploration and exploitation.
> > >
> > > Since "novel methods to continuous MCTS are relatively rare," we hope to present our contribution at ICML. We'll ensure the revised version is properly attributed to Mao et al. and add all experimental results.

---

### Decision · Program_Chairs · 2025-05-01

**Decision:**

Accept (poster)

**Comment:**

This paper considers MCTS in stochastic environments with continuous action and state spaces, and proposes Stochastic-Power-HOOT, which integrates a power mean as a value backup operator and polynomial exploration bonus.

The paper has mixed reviews. Generally speaking, reviewers think the paper is promising, but detected several issues:

* For a paper that builds heavily upon previous work, there is a worrying lack of proper references and attribution of results to the correct work. This makes aspects regarding novelty and framing of the paper not as crisp as they should be;

* Experiments are limited;

*  Even though most reviewers praise the theoretical analysis provided, there are some doubts regarding justification of a few design choices and clarity regarding the action branching behaviour.

The authors acknowledge these issues and promise to address them in a final version. However, the gap between the current version of the paper and where in it needs to be is wider than other papers I have overseen.